# Triangle Multiplication is All You Need for Biomolecular Structure Representations

**Jeffrey Ouyang-Zhang**[1,2] [*], **Pranav Murugan**[1], **Daniel J. Diaz**[2], **Gianluca Scarpellini**[1],
**Richard Strong Bowen**[1], **Nate Gruver**[1], **Adam Klivans**[2], **Philipp Krähenbühl**[2],
**Aleksandra Faust**[1], **Maruan Al-Shedivat**[1]
[1]Genesis Research    [2]UT Austin

## Abstract

AlphaFold has transformed protein structure prediction, but emerging applications such as virtual ligand screening, proteome-wide folding, and de novo binder design demand predictions at a massive scale, where runtime and memory costs become prohibitive. A major bottleneck lies in the Pairformer backbone of AlphaFold3-style models, which relies on computationally expensive triangular primitives—especially triangle attention—for pairwise reasoning. We introduce *Pairmixer*, a streamlined alternative that eliminates triangle attention while preserving higher-order geometric reasoning capabilities that are critical for structure prediction. *Pairmixer* substantially improves computational efficiency, matching state-of-the-art structure predictors across folding and docking benchmarks, delivering up to $4\times$ faster inference on long sequences while reducing training cost by 34%. Its efficiency alleviates the computational burden of downstream applications such as modeling large protein complexes, high-throughput ligand and binder screening, and hallucination-based design. Within BoltzDesign, for example, *Pairmixer* delivers over $2\times$ faster sampling and scales to sequences $\sim$30% longer than the memory limits of Pairformer. Code is available at https://github.com/genesistherapeutics/pairmixer.

## 1 Introduction

AlphaFold (Senior et al., 2020; Jumper et al., 2021) has transformed protein structure prediction and become an indispensable tool across the biological sciences. Yet emerging applications increasingly demand massive scale. Virtual screening of protein–ligand interactions, modeling of large protein complexes, proteome-wide folding, and iterative de novo binder design already require millions (and soon billions) of inference calls. At this scale, runtime and memory efficiency are critical bottlenecks: for example, Boltz-1 (Wohlwend et al., 2024) requires over 15 minutes to process a single 2048-token sequence on an A100 GPU (see Section 5.3). The dominant computational cost comes from pairwise token representations and triangular primitives, which scale *cubically* with sequence length $L$. While triangle multiplication is implemented efficiently via matrix multiplications, triangle attention requires $L$ attention operations, introducing substantial memory and runtime overhead.

We introduce *Pairmixer*, a streamlined alternative to the Pairformer backbone of AlphaFold3 (Abramson et al., 2024). By retaining triangle multiplication and feed-forward networks while eliminating triangle and sequence attentions, *Pairmixer* preserves the ability to reason over higher-order geometric interactions that are critical for structure prediction while alleviating Pairformer's heavy computational burden. Despite this simplification, *Pairmixer* performs comparably on RCSB and CASP15 test sets against state-of-the-art predictors such as AlphaFold, Chai-1, and Boltz-1, while providing $4\times$ faster inference on long sequences. *Pairmixer* consistently matches the performance of Pairformer backbone across protein-ligand, antibody-antigen, protein-nucleic acid and RNA structures while training in 34% fewer GPU-days across multiple model sizes (see Figure 1).

---

[*]Work done during an internship at Genesis Research
{jozhang,danny.diaz,klivans,philkr}@cs.utexas.edu
{pranav,gianscarpe,richard,ngruver,sandra,maruan}@genesistherapeutics.ai

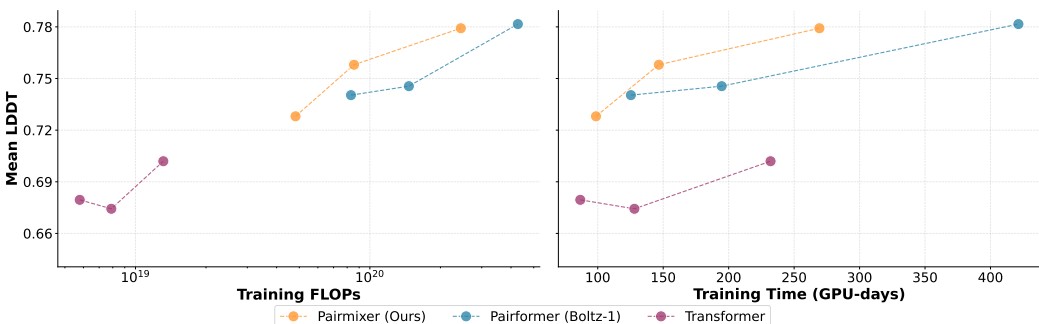

Figure 1: ***Pairmixer* is an efficient architecture for biomolecular structure prediction.** Across multiple model sizes, *Pairmixer* matches the performance of the leading Pairformer architecture while delivering greater training efficiency.

By reducing both runtime and memory requirements, *Pairmixer* expands the scope of feasible downstream applications of structure prediction. It enables modeling of larger protein complexes beyond the limits of triangle attention, supports high-throughput screening of ligands and binders, and accelerates hallucination-based design pipelines (Pacesa et al., 2025). Within the BoltzDesign1 (Cho et al., 2025) framework, *Pairmixer* provides over $2\times$ faster sampling and scales to sequences beyond 700 amino acids, where BoltzDesign otherwise fails due to memory overflow. Our analysis highlights the role of the pair representation in learning precise distances between residues and suggests that triangle multiplication learns to capture sparse long-range interactions among residue triplets.

## 2 RELATED WORK

**Biomolecular Structure Prediction.** Protein structure prediction has progressed rapidly in recent years, with early efforts primarily focused on modeling monomeric proteins (Senior et al., 2020; Jumper et al., 2021; Baek et al., 2021; Yang et al., 2020; Ahdritz et al., 2024). As these approaches matured, structure predictors expanded to handle multimeric assemblies (Evans et al., 2021; Baek et al., 2023) and other modalities such as nucleic acids (Baek et al., 2024). Today, state-of-the-art predictors can fold complexes that span a wide range of biomolecular types (Abramson et al., 2024; IntFold et al., 2025; Boitreaud et al., 2024; Wohlwend et al., 2024; ByteDance et al., 2025).

Biomolecular structure predictors rely on specialized backbones that capture complex geometric relationships among molecular entities. Early approaches such as trRosetta (Yang et al., 2020) and AlphaFold1 (Senior et al., 2020) leveraged convolutional neural networks to extract pairwise residue features from multiple sequence alignments (MSAs) and predict inter-residue distances. AlphaFold2 introduced the transformer-based Evoformer to jointly model MSA and pair representations (Jumper et al., 2021), while AlphaFold3 refined it with the Pairformer, which decouples MSA and pair processing (Abramson et al., 2024). The Pairformer has since become the de-facto backbone architecture for biomolecular structure prediction (IntFold et al., 2025; Boitreaud et al., 2024; Wohlwend et al., 2024; ByteDance et al., 2025). However, despite its strong performance, the Pairformer remains complex and computationally demanding.

Several alternative architectures have been proposed to simplify structure prediction backbones. Mini-Fold (Wohlwend et al., 2025) streamlines Alphafold2's Evoformer using a lightweight Miniformer based on triangle multiplications. SimpleFold (Wang et al., 2025) replaces the Evoformer with a sequence-only transformer that omits pair representations. Our work also simplifies backbone design, but unlike prior efforts focused on monomeric folding, *Pairmixer* is developed for AlphaFold3-like cofolding models, enabling structure prediction across broader biomolecular modalities.

**Downstream Applications of Structure Prediction.** The success of biomolecular structure prediction has enabled a growing number of downstream applications, many of which leverage predicted structures at unprecedented scales. Large-scale resources such as the AlphaFold Database (Varadi et al., 2022) and OpenFold (Ahdritz et al., 2024) have generated massive synthetic protein structure datasets using AlphaFold2, powering advances in structure search (Van Kempen et al., 2024), protein language modeling (Heinzinger et al., 2024; Ouyang-Zhang et al., 2024; Hayes et al., 2025), and diffusion-based structure generation (Geffner et al., 2025; Lin et al., 2024; Daras et al., 2025).

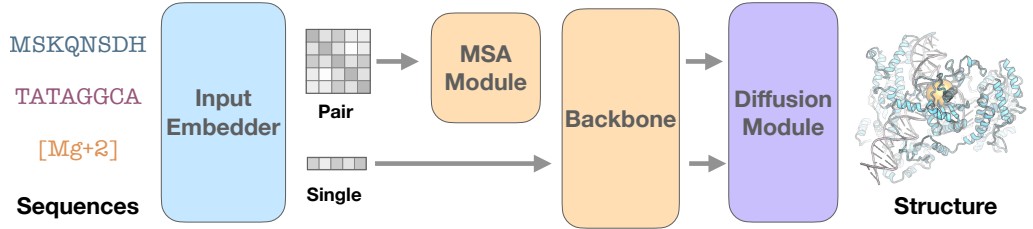

Figure 2: **Overview of Biomolecular Structure Prediction.** Given a list of sequences, our model predicts the 3D folded structure of all sequences within a single complex. Input sequences are first *embedded* into a single representation for each residue and a pair representation to capture the relationship between pairs of residues. The *MSA Module* and *Backbone* (e.g., Pairformer) extracts deep pairwise features capturing inter-residue interactions, which are then passed to the *diffusion module* to generate the 3D structure. (Additional inputs such as MSAs, conformers, and templates are omitted for clarity.)

Structure predictors now drive virtual screening pipelines that evaluate millions of candidate drugs based on predicted protein–ligand interactions (Wong et al., 2022; Shamir & London, 2025; Scardino et al., 2023), and large-scale folding studies that map the human interactome (Ille et al., 2025; Zhang et al., 2025). Hallucination-based generation methods such as BindCraft (Pacesa et al., 2025) further use predictors in iterative optimization loops requiring millions of model evaluations. As these applications expand in scope and scale, inference speed becomes a critical bottleneck. We introduce a structure predictor that matches state-of-the-art accuracy while operating at a fraction of the runtime, enabling faster and broader deployment of downstream workflows.

**Attention-free Architectures.** While transformers lead modern architectures, attention-free variants aim to improve scalability. FNet (Lee-Thorp et al., 2021) and related models (Poli et al., 2023; Zhai et al., 2021) replace attention with Fourier or convolutional mixing for sub-quadratic efficiency, while MLP-Mixer (Tolstikhin et al., 2021) achieves competitive performance using token- and channel-wise multi-layer perceptrons (MLPs). *Pairmixer* removes attention entirely from the backbone and mixes tokens through matrix multiplication.

Architectures based on triangle multiplication have been explored in several prior works. Genie2 (Lin et al., 2024) performs de-novo structure generation by iteratively updating a pair representation through triangle multiplications, while MSA Pairformer (Akiyama et al., 2025) applies similar operations to extract features from multiple sequence alignments. IgFold (Ruffolo et al., 2023) incorporates triangle operations within GNN layers. *Pairmixer* likewise learns rich protein representations through triangle multiplication, but in the context of biomolecular structure prediction.

## 3 PRELIMINARIES

Let $x = \{x^{(1)}, \cdots, x^{(K)}\}$ denote a collection of $K$ biomolecular sequences. Each sequence $x^{(k)} = (x_1^{(k)}, \cdots, x_{L^{(k)}}^{(k)})$ consists of tokens $x_i^{(k)} \in \mathcal{T}$ corresponding to an amino acid, a nucleic acid, or small molecule heavy atoms. $L^{(k)}$ denotes the number of tokens in biomolecule $x^{(k)}$. The goal of biomolecular structure prediction is to map the sequences $x$ to a three-dimensional structure $a = \{a^{(1)}, \cdots, a^{(K)}\}$, where each biomolecular structure $a^{(k)} = (\boldsymbol{a}_1^{(k)}, \cdots, \boldsymbol{a}_{N^{(k)}}^{(k)})$ consists of atomic coordinates $\boldsymbol{a}_j^{(k)} \in \mathbb{R}^3$, and $N^{(k)}$ denotes the number of atoms in biomolecule $k$. See Figure 2 for an overview.

**The Input Embedder** concatenates the sequences $x = \{x^{(1)}, \ldots, x^{(K)}\}$ and embeds it into a "*single*" length $L = \sum_{k=1}^{K} L^{(k)}$ sequence representation $\boldsymbol{s}^{\text{init}} \in \mathbb{R}^{L \times C_s}$ of dimension $C_s$. Modern structure predictors (Jumper et al., 2021) additionally initialize a "*pair*" representation $\boldsymbol{z}^{\text{init}} \in \mathbb{R}^{L \times L \times C_z}$:

$$\boldsymbol{z}_{ij} = \boldsymbol{s}_i + \boldsymbol{s}_j + \mathbf{PE}(i, j),$$

where $\mathbf{PE}(i, j)$ is a positional encoding that incorporates both intra- and inter-sequence distances and $C_z$ is the pair embedding dimension. Intuitively, $\boldsymbol{z}_{ij} \in \mathbb{R}^{C_z}$ captures the relational context between tokens $\boldsymbol{s}_i$ and $\boldsymbol{s}_j$ and enables the model to reason about longer-range couplings. Since pairwise reasoning is critical for structure prediction, we adopt the same input embedding scheme.

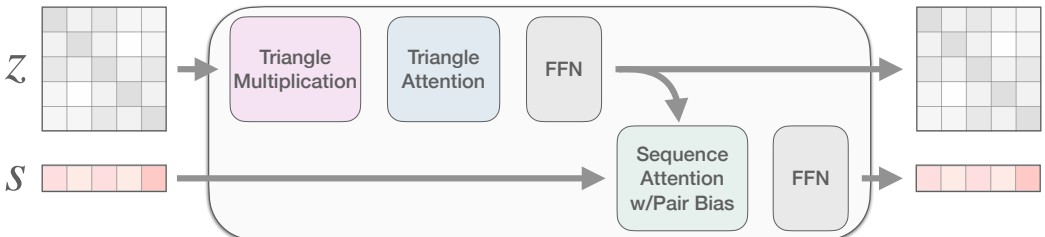

(a) **Pairformer architecture.** The de facto biomolecular structure prediction backbone.

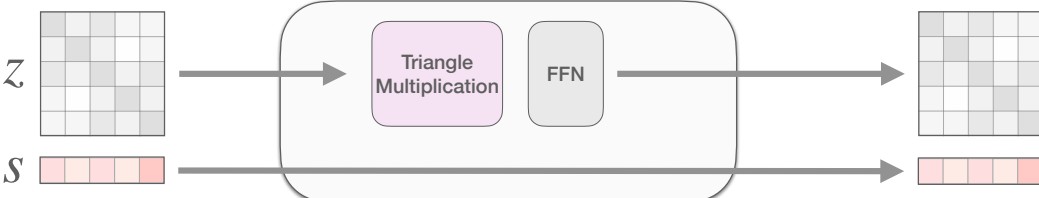

(b) *Pairmixer* **architecture.** An efficient yet effective biomolecular structure prediction backbone.

Figure 3: **Schematic comparison of the Pairformer and *Pairmixer* backbones.** *Pairmixer* simplifies the Pairformer architecture by removing redundancies. This results in faster training and inference, expanding the scale of downstream applications.

**The MSA Module** encodes evolutionary information that is crucial for structure prediction (Benner & Gerloff, 1991; Yanofsky et al., 1964; Ovchinnikov et al., 2017; 2014; Morcos et al., 2011; Weigt et al., 2009). For each amino acid or nucleic acid sequence $x^{(k)}$, we perform a homology search to construct a multiple sequence alignment (MSA) of related sequences that likely adopt the same fold. Formally, $\mathbf{MSA}(x^{(k)}) \in (\mathcal{T} \cup \{\texttt{GAP}\})^{M^{(K)} \times L^{(K)}}$ contains $M^{(k)}$ aligned sequences of length $L^{(k)}$. This alignment establishes positional correspondence across homologous sequences, enabling detection of conserved sites and co-evolutionary couplings. The resulting MSAs are then paired, concatenated, and embedded into $\boldsymbol{m}^{\mathrm{init}} \in \mathbb{R}^{M \times L \times C_m}$ where $M$ is the number of filtered homologous sequences and $C_m$ is the MSA embedding dimension.

The MSA module takes $(\boldsymbol{m}^{\mathrm{init}}, \boldsymbol{z}^{\mathrm{init}})$ as input, extracts structurally-relevant evolutionary patterns from $\boldsymbol{m}^{\mathrm{init}}$, and encodes pairwise interactions into $\boldsymbol{z}^{\mathrm{msa}}$ to guide folding. Since processing all $M$ sequences in the MSA is computationally expensive, AlphaFold3 introduced a shallow 4-layer MSA module after which the MSA is discarded while the evolutionary-aware pair representation $\boldsymbol{z}^{\mathrm{msa}}$ continues to be refined. Our model derives $\boldsymbol{z}^{\mathrm{msa}}$ from an MSA module but introduces a more efficient feature extractor to refine its evolutionary signals.

**The Pairformer backbone** serves as the primary feature extractor for AlphaFold3 (Abramson et al., 2024), producing structrually-aware representations that encode geometric constraints between residues (see Figure 3a). It takes $(\boldsymbol{s}^{\mathrm{init}}, \boldsymbol{z}^{\mathrm{msa}})$ as input and employs several specialized modules that iteratively update the sequence and pair representations to produce $(\boldsymbol{s}^{\mathrm{backbone}}, \boldsymbol{z}^{\mathrm{backbone}})$. See Figure 11 for a more detailed treatment of the entire architecture.

The Pairformer contains two specialized modules for processing the pair representation: triangle attention and triangle multiplication. These modules treat the pair representation $\boldsymbol{z} \in \mathbb{R}^{L \times L \times C_z}$ as edge features of a fully-connected graph of $L$ nodes and reason over triplets of residues (nodes) to learn geometric constraints.

*Triangle attention* computes attention (with pair bias) along every row (and column) of the pair representation. Formally, the update to row $i$ is

$$\mathbf{TriAtt}(\boldsymbol{z})_i = \mathrm{softmax}\Big((\boldsymbol{W}_Q \boldsymbol{z}_i)(\boldsymbol{W}_K \boldsymbol{z}_i)^\top + \boldsymbol{W}_B \boldsymbol{z}\Big) \boldsymbol{W}_V \boldsymbol{z}_i$$

where $(\boldsymbol{W}_Q, \boldsymbol{W}_K, \boldsymbol{W}_V)$ are standard attention projection matrices, and $\boldsymbol{W}_B$ projects the pair representation into an attention bias term [1]. $\mathbf{TriAtt}(\boldsymbol{z})_i$ effectively performs attention over all residues while conditioning on residue $i$.

---

[1] single head and removed scaling for brevity

---

**Algorithm 1** *Pairmixer* Backbone

---

**Require:** Input pair representation $\boldsymbol{z}^{\mathrm{msa}} \in \mathbb{R}^{L \times L \times C_z}$
**Require:** Number of backbone layers $N$
**Ensure:** Updated pair representation $\boldsymbol{z}_N$
 1:  $\boldsymbol{z}_0 \leftarrow \boldsymbol{z}^{\mathrm{msa}}$
 2:  **for** $l = 0$ to $N - 1$ **do**
 3:     $\boldsymbol{z}_l \leftarrow \boldsymbol{z}_l + \textbf{TriMulIncoming}(\boldsymbol{z}_l)$
 4:     $\boldsymbol{z}_l \leftarrow \boldsymbol{z}_l + \textbf{TriMulOutgoing}(\boldsymbol{z}_l)$
 5:     $\boldsymbol{z}_{l+1} \leftarrow \boldsymbol{z}_l + \textbf{FFN}(\boldsymbol{z}_l)$
 6:  **end for**
 7:  **return** $\boldsymbol{z}_N$

---

*Triangle multiplication* performs matrix multiplications to integrate features across different rows (and columns) of the pair representation. Formally, the update to edge $\boldsymbol{z}_{ij}$ is

$$\textbf{TriMul}(\boldsymbol{z})_{ij} = \sum_{k=1}^{L} (\boldsymbol{W}_a \boldsymbol{z}_{ik}) \odot (\boldsymbol{W}_b \boldsymbol{z}_{jk})$$

where $\boldsymbol{W}_a, \boldsymbol{W}_b$ are linear projection layers. For each edge $\boldsymbol{z}_{ij}$, triangle multiplication computes how every node $k$ interacts with query nodes $i$ and $j$ through edges $\boldsymbol{z}_{ik}$ and $\boldsymbol{z}_{jk}$.

Both operations scale cubically with sequence length, making the processing of long sequences computationally expensive. Triangle multiplication is more efficient, as it can be implemented with matrix multiplications (e.g., `torch.einsum`), whereas triangle attention incurs the higher cost of $L$ full attention computations. In this work, we streamline the cofolding backbone to its essential components and show that triangle multiplication yields representations as powerful as those from triangle attention, but at substantially lower computational cost, supporting a range of downstream applications.

While the Pairformer is trained with an auxiliary distogram loss that ensures $\boldsymbol{z}^{\mathrm{backbone}}$ accurately represents all pairwise token distances, it does not yet specify an atomic 3-D structure.

**The Diffusion Module** samples the atomic coordinates conditioned on $(\boldsymbol{s}^{\mathrm{backbone}}, \boldsymbol{z}^{\mathrm{backbone}})$. It uses transformers to derive atomic representations from the token-level sequence and pair representations, and subsequently denoises all-atom coordinates based on these representations. We leverage the diffusion module as-is to realizes 3-D structures conditioned on single and pair representations derived from our efficient backbone.

## 4  METHOD

We introduce *Pairmixer*, an attention-free feature extractor for biomolecular structure prediction and design (see Figure 3). *Pairmixer* exclusively updates the pair representation $\boldsymbol{z}^{\mathrm{msa}}$, leaving the single-sequence representation $\boldsymbol{s}^{\mathrm{init}}$ unchanged. Through *triangle multiplication*, *Pairmixer* efficiently mixes features within the pair representation, facilitating reasoning over residue triplets and their geometric constraints. Combined with feed-forward networks (FFN) that process all residue pairs, this architecture provides an effective and expressive backbone for biomolecular structure prediction.

The full algorithmic specification of *Pairmixer* is available in Algorithm 1. In developing *Pairmixer*, we identified and removed two unnecessary modules from the Pairformer: sequence updates and triangle attention.

**Removing Sequence Updates.** In AlphaFold2's Evoformer backbone, sequence updates were essential components that processed the MSA to capture evolutionary features. However, the MSA Module in cofolding models now preprocesses the MSA and encodes this evolutionary information directly into the pair representation $\boldsymbol{z}^{\mathrm{msa}}$, eliminating the need for sequence updates to provide evolutionary information. Since the pair updates proved more expressive, we bypass sequence processing entirely and pass the initial sequence representation directly to the diffusion module (i.e., $\boldsymbol{s}^{\mathrm{backbone}} = \boldsymbol{s}^{\mathrm{init}}$).

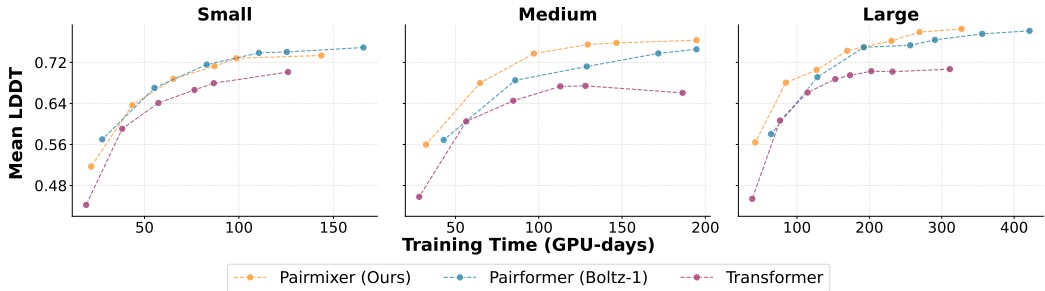

Figure 4: **Performance curves on RCSB test set across model sizes.** We compare three backbone architectures across three model sizes over training. *Pairmixer* matches or surpasses the Pairformer baseline while training more efficiently.

**Removing Triangle Attention.** Triangle attention reasons over residue triplets by applying attention to each row of the pair representation $z_i$, using the full $z$ as pairwise bias (see Figure 11b). However, this approach is computationally expensive, requiring $L$ separate attention operations over $L$ tokens per layer. Triangle multiplication offers equivalent capability for capturing geometrically consistent pair representations via a triplet reasoning mechanism, but with significantly lower computational cost. Since both methods have independently demonstrated strong performance in structure prediction (Jumper et al., 2021), we adopt the more efficient triangle multiplication approach.

## 5 RESULTS

### 5.1 IMPLEMENTATION DETAILS

We implement *Pairmixer* on top of Boltz-1, an *AlphaFold3* descendant. More specifically, we replace the Pairformer backbone with *Pairmixer* and remove triangle attention from the MSA Module. Note that we do not alter the diffusion module's transformer architecture. We also introduce a transformer baseline that preserves the sequence update while removing the pair update in the backbone. To ensure this baseline is as strong as possible, we modify the architecture to allow features to flow effectively from the MSA module into the diffusion module (see Section A.2).

Following Boltz-1 training schedule (Wohlwend et al., 2024), we train on 384/3456 token/atom crops for the first 53k iterations using the PDB and OpenFold distillation dataset. We then finetune for 15k iterations on the PDB dataset with a larger crop size of 512/4608. To evaluate the generality of our approach, we train models of multiple sizes. Our large configuration matches Boltz-1, with 48 Pairformer layers and 24 diffusion transformer layers. In addition, we develop small and medium variants with 12/24 Pairformer layers and 6/24 diffusion transformer layers, respectively. During inference, we default to 10 recycling steps and 200 sampling steps for all models. In our main evaluation, we sample 5 poses and report the metrics on the top pose (oracle evaluation). Full hyperparameter details are in Table 10.

### 5.2 COMPARISONS ON COFOLDING PERFORMANCE ACROSS MODEL SIZES

We evaluate our efficient *Pairmixer* architecture against two baselines, Pairformer (Abramson et al., 2024) and a sequence-only Transformer. All models are evaluated on the RCSB test set introduced in Boltz-1 (Wohlwend et al., 2024), which contains 533 structures with at most 40% sequence identity to the training set, maximum small-molecule similarity of 80%, and resolution better than 4.5Å. All models are evaluated at 15, 30, 45, 60, and 68 epochs, totalling 53k iterations and the large model is additionally evaluated during the second phase of 15k iterations. We additionally extend training for small and medium *Pairmixer* and Transformer models until the total training time matches the Pairformer. We report the final mean LDDT, averaged across all residues.

Our *Pairmixer* consistently outperforms or matches the Pairformer across all model sizes (see Figure 4). At the large scale, *Pairmixer* reaches Pairformer-level accuracy (mean LDDT of 0.78) while requiring only 66% of the training time. The trend holds at smaller scales: *Pairmixer* surpasses Pairformer at the medium scale and matches it at the small scale under equal training budgets. Fur-

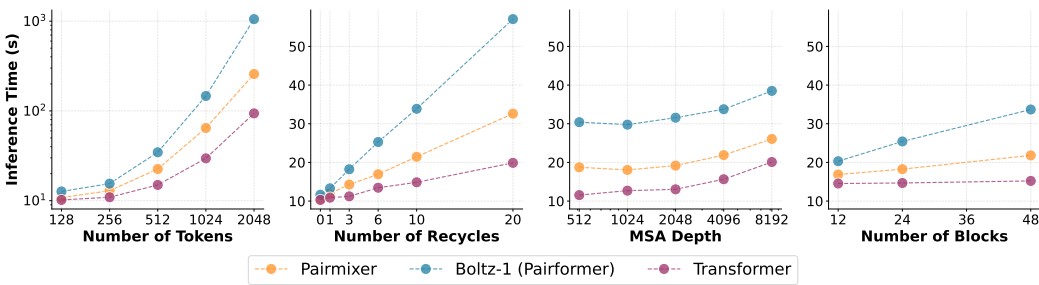

Figure 5: **Inference speed analysis.** We measure runtime across architectures and input sizes. While the Transformer is the fastest overall, *Pairmixer* achieves substantially lower inference times than Pairformer, particularly on longer sequences.

thermore, under the same training time, *Pairmixer* exceeds the sequence-only Transformer baseline across all scales. These results suggest that a sequence-only Transformer is inadequate for extracting structural features, while the triangle multiplications and feed-forward networks in *Pairmixer* are sufficient to capture rich structural representations. Full tabular results are provided in Table 4 and Table 5, and detailed FLOPs analysis is provided in Section B.

## 5.3 INFERENCE TIME COMPARISONS

Many downstream applications require running the structure predictor on thousands to millions of complexes, making inference efficiency critical. In Figure 4, we benchmark *Pairmixer* against the Pairformer and a sequence-only transformer under a default setting of 512 tokens, 4608 atoms, MSA depth of 4096, 10 recycles, 48 blocks, and 200 sampling steps.

On this setup, Boltz-1 requires 34 seconds to generate a single sample on a GH200 GPU, while *Pairmixer* completes in 21 seconds, yielding a 1.6× speedup. This advantage holds consistently across different recycle counts, MSA depths, and backbone sizes. The scaling benefits are even more striking for longer sequences: at 1024 tokens, *Pairmixer* is 2× faster, and at 2048 tokens, it delivers a 4× speedup, reducing runtime from 1000 seconds to 250 seconds. These results establish *Pairmixer* as a scalable and efficient architecture, making large-scale cofolding more practical.

## 5.4 COMPARISONS TO PRIOR WORKS

Figure 6 compares *Pairmixer* to other cofolding models on the RCSB test set, evaluating protein folding, protein–protein interactions (DockQ), and protein–ligand interactions (lDDT-PLI and ligand RMSD < 2). See Section 5.2 for a description of the test dataset. We generate five poses per complex and report both the performance of the best pose (oracle) and the average across poses. Results

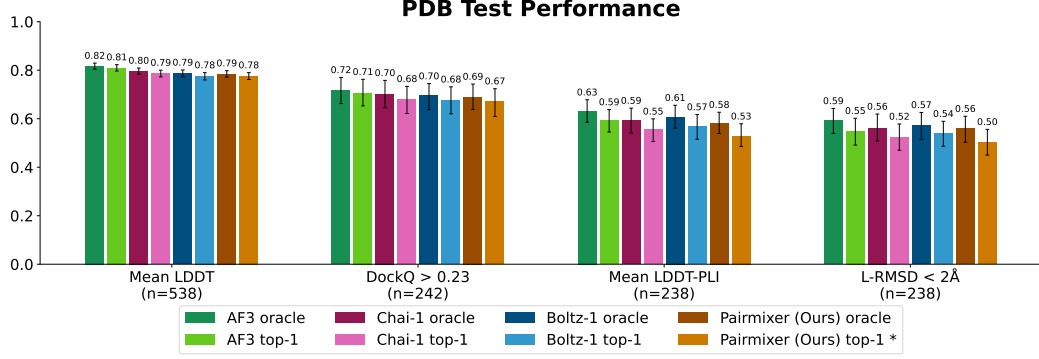

Figure 6: **System-level comparison on the RCSB test set.** We evaluate against AlphaFold3, Chai-1, and Boltz-1 on protein and small-molecule structure prediction. *Pairmixer* performs competitively with these state-of-the-art approaches. Error bars denote bootstrapped 95% confidence intervals. *Since we do not train a confidence model, results are reported using the first prediction.

Table 1: **Performance on diverse biomolecular structure prediction benchmarks.** Number of test samples for each dataset is indicated in parentheses.

(a) **PoseBusters**: Protein-Ligand Complex (298)

| Method | $RMSD_{<2}$ | $RMSD_{<1}$ | $lDDT_{PLI}$ |
|---|---|---|---|
| Pairformer (Boltz-1) | 0.68 | 0.46 | 0.74 |
| Pairmixer (Ours) | 0.67 | 0.45 | 0.73 |

(b) **Antibody–Antigen Complex** (70)

| Method | $DOCKQ_{>0.23}$ |
|---|---|
| Pairformer (Boltz-1) | 0.23 |
| Pairmixer (Ours) | 0.23 |
| Transformer | 0.08 |

(c) **Protein–Nucleic Acid Complex** (172)

| Method | ICS | IPS |
|---|---|---|
| Pairformer (Boltz-1) | 0.50 | 0.65 |
| Pairmixer (Ours) | 0.51 | 0.66 |
| Transformer | 0.48 | 0.64 |

(d) **RNA Structure** (27)

| Method | lDDT |
|---|---|
| Pairformer (Boltz-1) | 0.58 |
| Pairmixer (Ours) | 0.59 |
| Transformer | 0.61 |

for existing methods are taken from the literature. *Pairmixer* matches Boltz-1 in mean lDDT and protein–ligand lDDT, slightly improves ligand RMSD < 2 (0.55 vs. 0.54), but lags on DockQ > 0.23 (0.63 vs. 0.64). These results indicate that even at the largest scale, triangle multiplication and pair FFNs in *Pairmixer* are sufficient for cofolding across diverse interaction types. We show similar results on the CASP15 test set in Section C.1.

## 5.5 COMPARISONS ON DIVERSE STRUCTURE PREDICTION TASKS

Table 1 shows evaluation results across a variety of biomolecular structure prediction benchmarks, including protein–ligand complexes (PoseBusters), antibody–antigen complexes, protein–nucleic acid complexes, and RNA structures. Experimental details are provided in Section C.3. Pairmixer performs on par with Pairformer, while standard Transformers generally lag behind. The exception is RNA structures, where the Transformer baseline slightly outperforms both Pairformer and Pairmixer, likely due to the limited availability of RNA structural training data. Notably, Pairmixer achieves comparable performance to Pairformer despite not using sequence attention. These results highlight Pairmixer's generality and robustness in modeling diverse biomolecular interactions.

## 5.6 COMPARISONS ON BINDER DESIGN (BINDFAST)

Hallucination-based protein design methods have shown that structure predictors can act as differentiable scoring functions for sequence optimization. However, they are memory-intensive and slow, requiring hundreds of runs to generate a single sequence. We introduce BindFast, which replaces BoltzDesign's (Cho et al., 2025) Pairformer backbone with *Pairmixer*, reducing runtime and memory usage. On 80GB A100 GPU, BoltzDesign encountered OOM errors on targets with over 500 residues, while BindFast handled targets up to 650 residues (+30%) and ran over $2\times$ speedups (see Table 2). Qualitative comparisons in Figure 13 show comparable designs, suggesting BindFast enables faster in-silico iteration and design of larger, biologically relevant binders. Details are in Section C.4.

Table 2: **Runtime comparison of generating proteins with *Pairmixer* and Pairformer in the BoltzDesign framework.** For biologically relevant targets of various sequence lengths, we generate three 110-residue binders using 160 iterations in all settings and report the average running time.

| Target | PDB_Chain | Complex Length | Target Length | Pairformer Time (sec) | Pairmixer Time (sec) | Speedup |
|---|---|---|---|---|---|---|
| GIP peptide | 2QHK_B | 140 | 30 | 680 | 337 | 2.01× |
| Ubiquitin | 1UBQ_A | 186 | 76 | 1113 | 532 | 2.09× |
| TP53 | 4MZI_A | 303 | 193 | 3198 | 1390 | 2.30× |
| hSDH | 1P5J_A | 429 | 319 | 7289 | 2920 | 2.50× |
| hMAO | 1GOS_A | 607 | 497 | 17134 | 6601 | 2.60× |
| bsDNA Polymerase | 3TAN_A | 702 | 592 | OOM | 9184 | ∞ |
| hTLR3 | 1ZIW_A | 739 | 629 | OOM | 10568 | ∞ |
| Prostate Antigen (PSA) | 1Z8L_A | 805 | 695 | OOM | OOM | – |

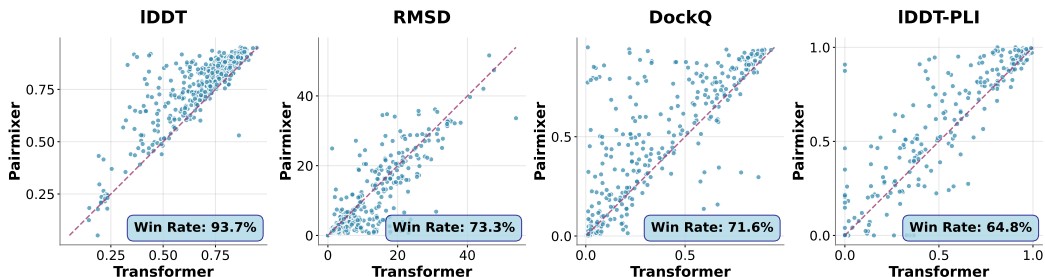

Figure 7: **Head-to-head comparison between *Pairmixer* and the Transformer backbone.** The win rate shows how often the *Pairmixer* architecture achieves a better score than the Transformer architecture. *Pairmixer* outperforms the Transformer on the distance-based lDDT metric in 93.7% of the cases, highlighting that its advantage lies in capturing pairwise interactions.

## 6 ANALYSIS

Predicting biomolecular structure requires reasoning over the entire sequence to capture diverse interactions among residues. We analyze how the architectural design of modern structure predictors facilitates such reasoning and how the simplified *Pairmixer* architecture achieves this.

**Pair representations.** A central challenge in biomolecular structure prediction is determining the strength of the interactions between all residue pairs. This is difficult because folding involves nonlocal tertiary interactions in which residues distant in sequence often interact physically in three-dimensional space. Modern structure predictors address this challenge with a pair representation. Our results indicate that the pair representation enables the model to capture fine-grain spatial relationships between all residue pairs.

We compare the performance of *Pairmixer*, which incorporates pair representations, against our sequence-only Transformer baseline in Figure 7. On the lDDT metric computed from pairwise distances, *Pairmixer* achieves higher scores in 93.7% of test complexes. In contrast, on the RMSD metric, which requires global structural alignment, the improvement is smaller (74.7%). These findings show that *pair representations provide greater benefits for local, pairwise accuracy* over sequence attention, suggesting their effectiveness in capturing residue–residue interactions.

**Triangle multiplication.** Modern structure predictors employ triangle attention and triangle multiplication within the pair representation to capture geometric relationships among residue triplets. While triangle attention allows the model to reason *sparsely* over interacting residues, triangle multiplication *densely* aggregates features across the entire sequence. However, our analysis shows that triangle multiplication also efficiently captures sparse geometric relationships among residue triplets by adjusting the magnitudes in the pair representations.

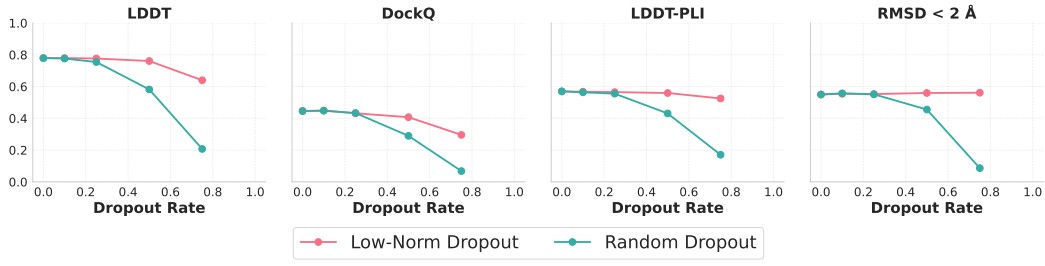

Figure 8: *Pairmixer* **Performance under different sparse triangle multiplication variants.** The model is trained with standard triangle multiplication and evaluated under various dropout conditions. While performance degrades rapidly under random dropout, it remains stable when low-norm entries in the triangle multiplication are zeroed out.

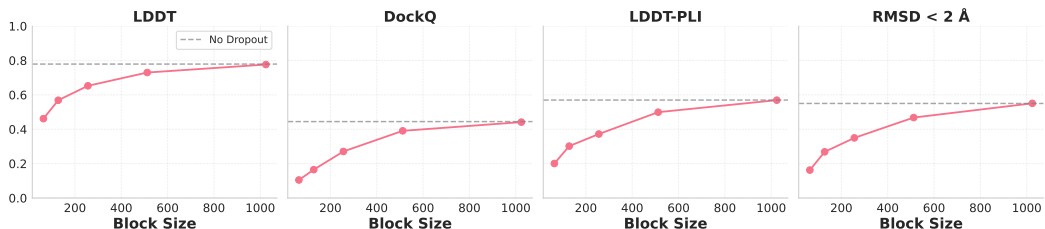

Figure 9: *Pairmixer* **Performance under blockwise dropout.** The model is trained with standard triangle multiplication and evaluated under a local blockwise triangle multiplication. Performance quickly degrades even for local metrics like lDDT.

We explicitly sparsify triangle multiplication by introducing dropout during inference. Formally,

$$\textbf{TriMulWithDropout}(\boldsymbol{z})_{ij} = \sum_{k=1}^{L} (\boldsymbol{W}_a \boldsymbol{z}_{ik}) \odot (\boldsymbol{W}_b \boldsymbol{z}_{jk}) \cdot \underbrace{M(\boldsymbol{z}_{ik}) \, M(\boldsymbol{z}_{jk})}_{\text{new dropout masks}}$$

where $M(\boldsymbol{z}_{ij}) \in \{0, 1\}$ determines whether a particular interaction is active.

In *random dropout* with dropout rate $\gamma \in [0, 1]$, the masks are sampled independently as $M(\boldsymbol{z}_{ik}), M(\boldsymbol{z}_{jk}) \sim \text{Bernoulli}(1 - \gamma)$. We experiment with a *low-norm dropout* scheme, dropping any interaction $(i, j)$ whose pair representation lies in the $\gamma \in [0, 1]$ fraction of smallest magnitudes. Formally, $M(\boldsymbol{z}_{ik}) = \begin{cases} 1, & \text{if } k \in \text{Top}_{1-\gamma}(\{\|\boldsymbol{z}_{il}\|\}_{l=1}^{L}) \\ 0, & \text{otherwise} \end{cases}$. Under both dropout schemes, each term $(\boldsymbol{W}_a \boldsymbol{z}_{ik}) \odot (\boldsymbol{W}_b \boldsymbol{z}_{jk})$ is retained only if both corresponding masks $M(\boldsymbol{z}_{ik})$ and $M(\boldsymbol{z}_{jk})$ are active, resulting in a higher effective dropout rate.

Figure 8 shows the performance of the model where both dropout schemes are applied to every layer with $\gamma = 0, 0.10, 0.25, 0.50, 0.75$. We observe that performance starts to degrade rapidly once the random dropout rate exceeds 25%, indicating that the model is not robust to random removal of interactions. However, the performance is very similar under the low-norm dropout of 75%. This suggests that, like attention, triangle multiplication identifies and processes a small subset of interactions that are essential for accurate folding of biomolecular complexes.

To probe which interactions the model relies on in its sparse computation, we evaluate it using a local block-dropout scheme. For block size $B$, we retain only local interactions: $M(\boldsymbol{z}_{ik}) = \begin{cases} 1, & \text{if } |i - k| \leq B \\ 0, & \text{otherwise} \end{cases}$. The results in Figure 9 show that performance already begins to degrade $B = 512$, with a substantial drop at $B = 256$ tokens. This suggests that *triangle multiplication processes sparse, long-range interactions*.

## 7 CONCLUSION

We introduce *Pairmixer*, a simplified, efficient feature extractor for biomolecular structure prediction. Models using *Pairmixer* train $1.5\times$ faster and sample up to $4\times$ faster than those with Pairformer, enabling large-scale, compute-intensive applications of structure prediction. The key idea is to explicitly materialize a 2-D pair representation, updated via triangle multiplications that capture interactions among residue triplets. We hypothesize that transforming 1-D sequences into 3-D structures is most effective when mediated through this intermediate pair representation, which naturally encodes distance information. Triangle multiplication provides a simple and efficient mechanism to do so.

## 8 ACKNOWLEDGMENT

This work was in part supported by the NSF AI Institute for Foundations of Machine Learning (IFML) and UT-Austin Center for Generative AI. The authors thank Yue Zhao, Sergey Ovchinnikov, Chengyue Gong and Luca Naef for their thoughtful feedback on this manuscript.

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

# A   ARCHITECTURAL BASELINES

The full cofolding pipeline for all methods can be found at Figure 10.

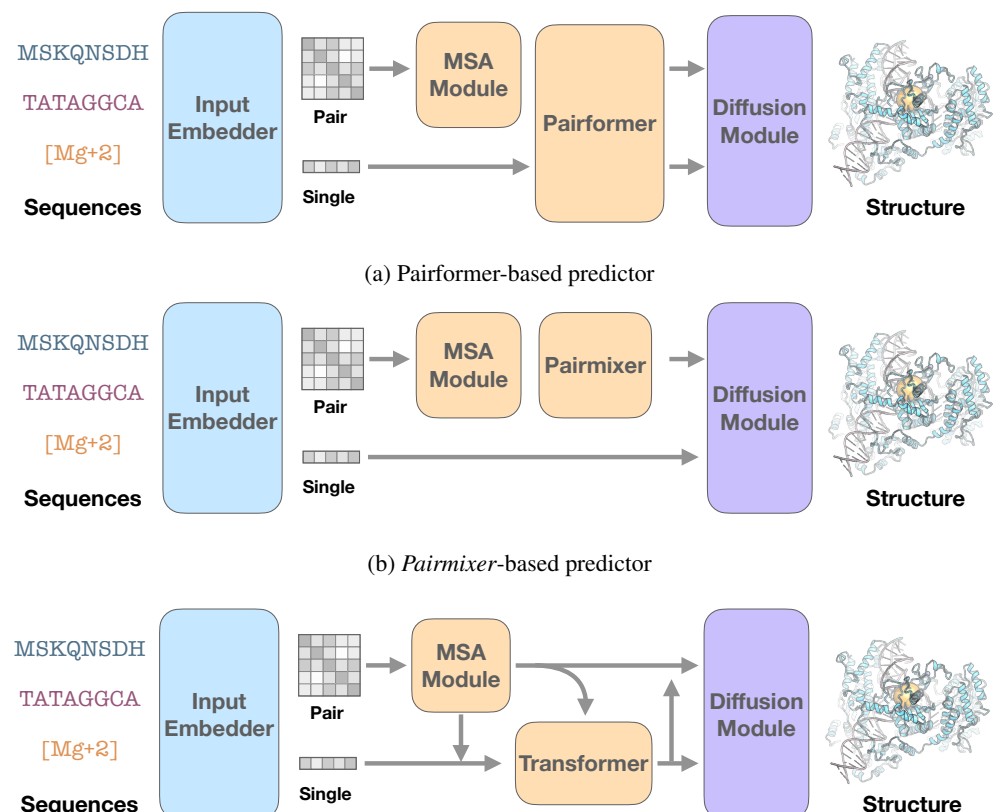

(a) Pairformer-based predictor

(b) *Pairmixer*-based predictor

(c) Transformer-based predictor

Figure 10: **Overview of biomolecular structure predictors.** We study the effect of varying backbone architectures while keeping all other modules fixed, except in the Transformer model, where we adjust the connections between the MSA module outputs and the Diffusion module inputs.

## A.1   PAIRFORMER BASELINE

Here we describe the Pairformer architecture of Figure 11 in detail.

**Attention Primitive.** The Pairformer extends the standard attention mechanism by incorporating a pairwise bias term derived from the pair representation $z$. Formally, this update is

$$\mathbf{AttnWithPairBias}(x, z) = \text{softmax}\Big((W_Q x)(W_K x)^\top + W_B z\Big) W_V x,$$

where $x \in \mathbb{R}^{L \times C_x}$ is a sequence representation, $z \in \mathbb{R}^{L \times L \times C_z}$ is a pair representation, $(W_Q, W_K, W_V)$ are standard attention projection matrices, and $W_B$ projects the pair representation into an attention bias term [2].

**The Sequence Update** first performs attention with pair bias (see Figure 11a) and then applies a feed-forward network. At layer $l$, we compute the update

$$\tilde{s}_{l+1} = s_l + \mathbf{AttnWithPairBias}(s_l, z_l)$$
$$s_{l+1} = \tilde{s}_{l+1} + \mathbf{FFN}(\tilde{s}_{l+1})$$

[2] single head and removed scaling for brevity

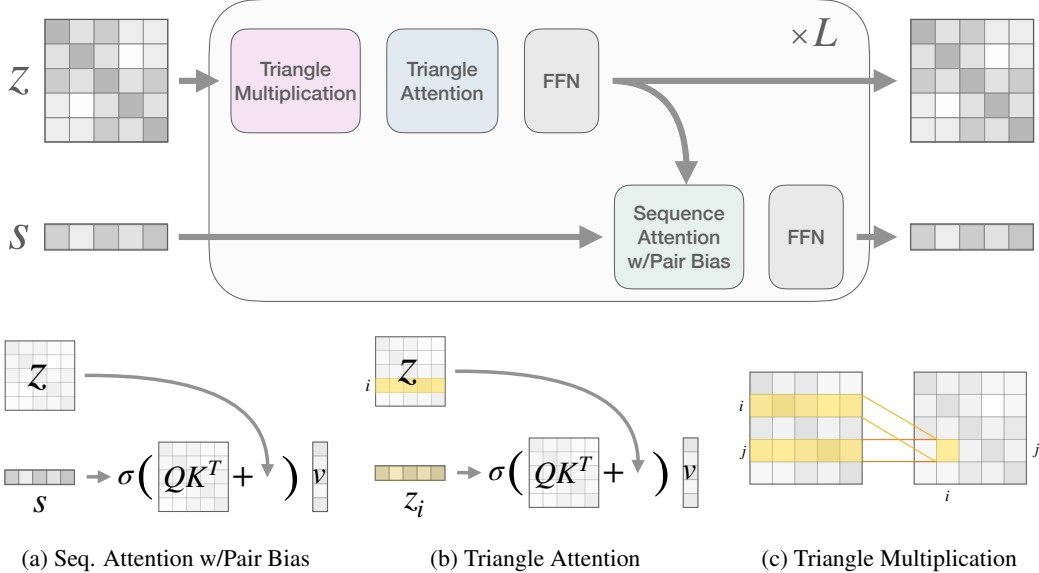

(a) Seq. Attention w/Pair Bias      (b) Triangle Attention      (c) Triangle Multiplication

Figure 11: **Pairformer Architecture and Module Details.** The main architecture (top) outlines the general Pairformer layer. The detailed module architectures (bottom) illustrate the key components: (a) Sequence Attention with Pair Bias, (b) Triangle Attention, and (c) Triangle Multiplication modules.

**The Pair Update** mixes the tokens in pair representation $z \in \mathbb{R}^{L \times L \times C_z}$ using triangle attention and triangle multiplication, then applies a feedforward network.

The *Triangle Attention* operates on each row of the pair representation $z_i \in \mathbb{R}^{L \times C_z}$ as an independent sequence, applying sequence attention with pair bias to each row separately[3] (see Figure 11b). Formally, the update for row $i$ is defined as

$$\mathbf{TriAttn}(z)_i = \mathbf{AttnWithPairBias}(z_i, z)$$

The *Triangle Multiplication* integrates features across different rows of the pair representation [4] (see Figure 11c). Formally, the update for feature $z_{ij}$ is defined as

$$\mathbf{TriMul}(z)_{ij} = \sum_{k=1}^{L} (W_a z_{ik}) \odot (W_b z_{jk})$$

where $W_a, W_b$ are linear projection layers.

Both pair operations were introduced to reason over triplets of residues, intuitively enabling the model to learn to follow geometric constraints in 3-D space (Jumper et al., 2021).

## A.2 TRANSFORMER BASELINE

Our transformer baseline removes the pair update from the Pairformer and keeps only the sequence update. We also modify the MSA module to make it more effective with the transformer baseline. Instead of outputting only $z^{\mathrm{msa}}$, it produces an additional sequence representation $s^{\mathrm{msa}}$, obtained by indexing the first row of the processed MSA representation. This $s^{\mathrm{msa}}$ is fed into the transformer, while $z^{\mathrm{msa}}$ serves as the pair bias. Additionally, the diffusion module expects both sequence and pair representations. Because the pair features are otherwise less processed in this baseline, we update them with the outer sum of the sequence representation. Formally,

$$z_{ij}^{\mathrm{backbone}} = z_{ij}^{\mathrm{msa}} + W_{s \to z} s_i^{\mathrm{backbone}} + W_{s \to z} s_j^{\mathrm{backbone}}$$

where $W_{s \to z} \in \mathbb{R}^{C_z \times C_s}$ is a projection layer. This is illustrated in Figure 10c.

---

[3]In practice, another layer of triangle attention is performed on the columns.

[4]In practice, another layer of triangle multiplication is performed on the columns.

| Module / Operation | FLOPs |
|---|---:|
| **Backbone / MSA Module** | |
| **Pair Update** | |
| **Triangle Attention** | |
| Matrix Multiply | $8\,L^3 C_z$ |
| Projection | $20\,L^2 C_z^2$ |
| **Triangle Multiplication** | |
| EinSum | $4\,L^3 C_z$ |
| Projection | $24\,L^2 C_z^2$ |
| **Pair FFN** | $24\,L^2 C_z^2$ |
| **Sequence Update** | |
| **Sequence Attention (with Pair Bias)** | |
| Matrix Multiply | $4\,L^2 C_s$ |
| Projection | $10\,L C_s^2$ |
| **Sequence FFN** | $24\,L C_s^2$ |
| **Diffusion Transformer** | |
| Attention (Pair Bias) – Matrix Multiply | $4\,L^2 C_a$ |
| Attention (Pair Bias) – Projection | $10\,L C_a^2$ |
| Sequence FFN | $16\,L C_a^2$ |
| **Full Modules** | |
| MSA Module | $R\,D_m\left(12L^3 C_z + 68L^2 C_z^2\right)$ |
| Pairformer | $R\,D_p\left(12L^3 C_z + 68L^2 C_z^2 + 4L^2 C_s + 34L C_s^2\right)$ |
| Structure Module | $M\,D_d\left(4L^2 C_a + 26L C_a^2\right)$ |

Table 3: **Breakdown of FLOPs in *AlphaFold3* architectural components.** Variables: $L = $ `max_tokens`, $C_z = $ `token_z`, $C_s = $ `token_s`, $C_a = 2 \times$ `token_z`, $R = $ `recycles`, $D_p = $ `pairformer_depth`, $D_m = $ `msa_depth`, $D_d = $ `diffusion_depth`, $M = $ `multiplicity`.

## B   FLOPs Calculations

Our biomolecular structure predictor uses a multi-resolution transformer that denoises atom coordinates at both the token and heavy-atom levels (see Figure 2). In this design, a backbone refines token representations, which are then processed by a conditional diffusion transformer. The backbone runs once per sequence, while the diffusion transformer can generate arbitrarily many samples.

In Table 3, we present the mathematical FLOP calculations for each component, and in Table 10 we report the total training and inference FLOPs for all model architectures.

**Boltz-1 Hyperparameters** The Boltz-1 architecture is defined by several key components and hyperparameters that influence its performance. We identify the following set of critical hyperparameters:

- **Input**: The input is defined by the number of input tokens ($L$), the single token dimension ($C_s$), and the pair token dimension ($C_z$).

- **Feature extractor**: The feature extractor consists of Pairformer and MSA blocks that process single and pair representations; its configuration is determined by the number of Pairformer blocks $D_p$, MSA blocks $D_m$.

- **Diffusion model**: The diffusion model is a transformer architecture made up of Multi-Head Attention (MHA) transformer layers. Its configuration is determined by the number of diffusion blocks ($D_d$) and the widths of its layers $C_a = 2\,C_z$.

**Feature extractors.** The feature extractors is a concatenation of $D_m$ MSA blocks and $D_p$ pairformer blocks. Each pairformer block primarily consists of two parallel update paths: the pair representation path and the single representation path (see Figure 11). Each path is further processed by a FFN. The pair representation path includes two *triangular self-attention* updates and two *triangular multiplication* updates (applied row-wise and column-wise). These are analogous to axial attention

mechanisms (Ho et al., 2019) operating over an $L \times L$ pair matrix, where each attention pass involves computations along one length-$L$ dimension for each of the $L$ rows or columns.

Each pair of triangular attention pass incurs a computational cost of $O(8L^3C_z)$ FLOPs. The triangle multiplication einsum operations require a quadratic FLOPs term per input token (total FLOPs of $4L^3C_z$). Following the triangle updates, a feed-forward network (FFN) is applied to each pair representation entry. The single representation path also contributes to the computational load, but its cost is quadratic in $L$.

Each MSA block is lighter than the full pairformer blocks and consists of a pair of triangular attention layers and a pair of triangular operations, followed by a FFN network for pair representation FFN ($C_z$), but without a single representation FFN and attention with pair bias. It also includes an additional OuterProductMean and pair-weighted averaging on the MSA, which we omit from our FLOPs calculations.

**Diffusion Model.** Each diffusion module block resembles a standard transformer block with a standard *self-attention* mechanism and a conditioning block. As with the trunk block analysis, we ignore bias terms, gating, and layer normalization for simplicity. We also ignore the cost of Atom Attention Encoder and Atom Attention Decoder that run on atoms, since those modules adopt sequence-local attention (Wohlwend et al., 2024) and their computational cost is negligible. The conditioned transition block of the diffusion model is dominated by dense matrix multiplications that scale quadratically with the hidden size $C_a$. The bulk of the compute arises from the SwiGLU feed-forward pathway, which contributes both a pair of linear projections ($4C_a^2$) and the associated activation matmul ($2C_a^2$). In addition, cross–path transformations are introduced via the $a \rightarrow b$ and $b \rightarrow a$ projections (each $2C_a^2$), followed by an output projection ($2C_a^2$). Finally, the gating mechanisms for both the $a$ and $b$ streams contribute another $2C_a^2$ apiece. The total FLOPs per structure block can therefore be approximated as the sum of the attention, MatMuls, and feed-forward components (see Table 3).

## C ADDITIONAL RESULTS

### C.1 SYSTEM-LEVEL COMPARISONS ON THE CASP15 DATASET

We report results on the CASP15 dataset in Table 12. These numbers differ slightly from Table 5 because we further filter proteins to ensure all methods are evaluated on the same set.

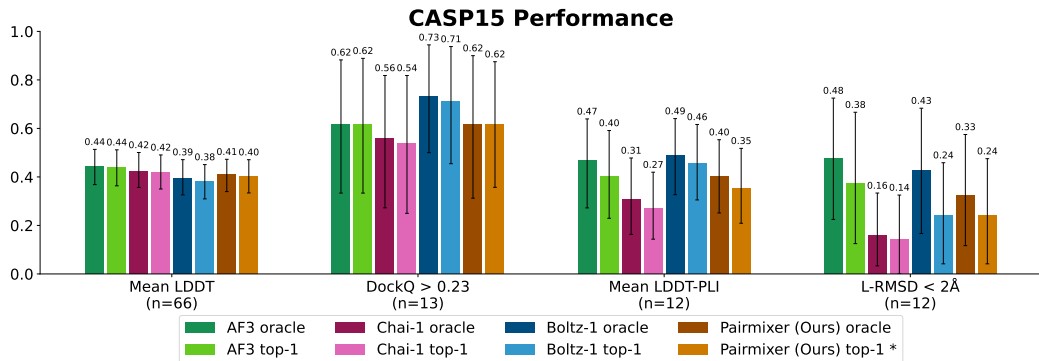

Figure 12: **System-level comparison on the CASP15 test set.** We evaluate against AlphaFold3, Chai-1, and Boltz-1 on protein and small-molecule structure prediction. *Pairmixer* performs competitively with these state-of-the-art approaches. Error bars denote bootstrapped 95% confidence intervals. *Since we do not train a confidence model, results are reported using the first prediction.

## C.2 FULL BIOMOLECULAR STRUCTURE PREDICTION RESULTS FOR RCSB AND CASP15

Table 4 and Table 5 report the full set of evaluation metrics across all architectures, along with the number of complexes evaluated by each metric. We retrain Boltz-1 for our Pairformer baselines and additionally include comparisons against the public checkpoint.

Table 4: **Model Performance on the Boltz RCSB test set.** The metric is computed on the best-performing protein out of five samples (oracle).

| Architecture | Epoch | GPU-Days | lDDT (n=539) | DOCKQ$_{>0.23}$ (n=342) | DOCKQ$_{>0.49}$ (n=342) | lDDT$_{PLI}$ (n=250) | RMSD$_{<1}$ (n=250) | RMSD$_{<2}$ (n=250) |
|---|---|---|---|---|---|---|---|---|
| **Small** | | | | | | | | |
| Transformer | 68 | 86 | 0.68 | 0.51 | 0.35 | 0.47 | 0.32 | 0.43 |
| Pairformer (Boltz-1) | 68 | 125 | 0.74 | 0.58 | 0.44 | 0.52 | 0.37 | 0.48 |
| Pairmixer (Ours) | 68 | 98 | 0.73 | 0.59 | 0.44 | 0.51 | 0.33 | 0.45 |
| **Medium** | | | | | | | | |
| Transformer | 68 | 128 | 0.67 | 0.50 | 0.36 | 0.47 | 0.33 | 0.46 |
| Pairformer (Boltz-1) | 68 | 194 | 0.75 | 0.60 | 0.47 | 0.53 | 0.36 | 0.49 |
| Pairmixer (Ours) | 68 | 146 | 0.76 | 0.60 | 0.46 | 0.54 | 0.40 | 0.53 |
| **Large** | | | | | | | | |
| Transformer | 68 | 173 | 0.69 | 0.51 | 0.37 | 0.48 | 0.33 | 0.46 |
| Pairformer (Boltz-1) | 68 | 290 | 0.76 | 0.61 | 0.49 | 0.54 | 0.41 | 0.52 |
| Pairmixer (Ours) | 68 | 192 | 0.75 | 0.61 | 0.46 | 0.55 | 0.38 | 0.51 |
| **Large Phase 2** | | | | | | | | |
| Transformer | 20 | 232 | 0.70 | 0.53 | 0.38 | 0.51 | 0.35 | 0.48 |
| Pairformer (Boltz-1) | 20 | 421 | 0.78 | 0.64 | 0.50 | 0.57 | 0.44 | 0.54 |
| Pairmixer (Ours) | 20 | 269 | 0.78 | 0.63 | 0.49 | 0.57 | 0.45 | 0.55 |
| **Boltz-1 public model** | | | | | | | | |
| Pairformer (Boltz-1) | - | - | 0.79 | 0.64 | 0.51 | 0.58 | 0.46 | 0.57 |

Table 5: **Model Performance on CASP15 test set.** The metric is computed on the best-performing protein out of five samples (oracle).

| Architecture | Epoch | GPU-Days | lDDT (n=66) | DOCKQ$_{>0.23}$ (n=14) | DOCKQ$_{>0.49}$ (n=14) | lDDT$_{PLI}$ (n=12) | RMSD$_{<1}$ (n=12) | RMSD$_{<2}$ (n=12) |
|---|---|---|---|---|---|---|---|---|
| **Small** | | | | | | | | |
| Transformer | 68 | 86 | 0.35 | 0.22 | 0.17 | 0.21 | 0.06 | 0.10 |
| Pairformer (Boltz-1) | 68 | 125 | 0.39 | 0.46 | 0.24 | 0.36 | 0.10 | 0.21 |
| Pairmixer (Ours) | 68 | 98 | 0.37 | 0.39 | 0.21 | 0.35 | 0.06 | 0.16 |
| **Medium** | | | | | | | | |
| Transformer | 68 | 128 | 0.35 | 0.19 | 0.16 | 0.27 | 0.04 | 0.15 |
| Pairformer (Boltz-1) | 68 | 194 | 0.38 | 0.66 | 0.35 | 0.39 | 0.14 | 0.23 |
| Pairmixer (Ours) | 68 | 146 | 0.39 | 0.49 | 0.39 | 0.38 | 0.12 | 0.24 |
| **Large** | | | | | | | | |
| Transformer | 68 | 173 | 0.36 | 0.29 | 0.16 | 0.26 | 0.06 | 0.10 |
| Pairformer (Boltz-1) | 68 | 290 | 0.41 | 0.68 | 0.43 | 0.37 | 0.12 | 0.31 |
| Pairmixer (Ours) | 68 | 192 | 0.38 | 0.50 | 0.35 | 0.34 | 0.12 | 0.23 |
| **Large Phase 2** | | | | | | | | |
| Transformer | 20 | 232 | 0.37 | 0.34 | 0.17 | 0.26 | 0.11 | 0.11 |
| Pairformer (Boltz-1) | 20 | 421 | 0.42 | 0.64 | 0.43 | 0.36 | 0.10 | 0.28 |
| Pairmixer (Ours) | 20 | 269 | 0.41 | 0.52 | 0.36 | 0.34 | 0.14 | 0.31 |
| **Boltz-1 public model** | | | | | | | | |
| Pairformer (Boltz-1) | - | - | 0.4 | 0.68 | 0.43 | 0.45 | 0.23 | 0.42 |

## C.3 DETAILS FOR DIVERSE BIOMOLECULAR STRUCTURE PREDICTION

We evaluate *Pairmixer* across several benchmark datasets listed in Table 1. Below, we describe the dataset preparation and evaluation protocols used for these benchmarks.

**Protein-ligand complexes.** We evaluate performance on protein–ligand complexes using the Pose-Buster benchmark. The original dataset contains 428 complexes. Applying a training-date cutoff of September 30, 2021 reduces this to 373, and after removing redundant protein–ligand complexes, the final benchmark includes 298 structures. Evaluation uses standard protein–ligand metrics, including RMSD < 2Å, RMSD < 1Å, and protein–ligand lDDT. We compare a Pairmixer model finetuned for longer to the publicly available Boltz-1 checkpoint with Pairformer in Table 1a. Under this setup, *Pairmixer* performs comparably to Pairformer, with at most a 1% drop in performance, while the architecture is significantly simpler and more efficient, requiring no attention in the backbone.

**Antibody–antigen complexes.** We evaluate on the antibody–antigen benchmark introduced in the AlphaFold3 paper. First, we extract the relevant chains from the publicly released AlphaFold3

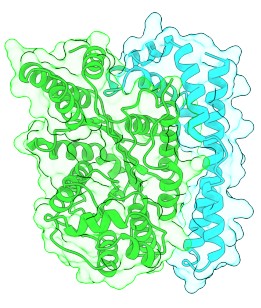
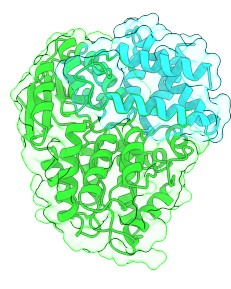

(a) Pairformer-based predictions                    (b) *Pairmixer*-based predictions

Figure 13: **Qualitative visualizations of de-novo binders.** Target is shown in green and binder is shown in blue. PDB code: 1P5J

files and remove unresolved residues. Of the 71 total complexes, 70 pass our data pipeline. The protein-protein interface is evaluated using the DockQ > 0.23 metric. We compare large Pairformer, Pairmixer, and Transformer models trained for the same number of iterations in Table 1b. We find that *Pairmixer* matches the performance of Pairformer (0.23), while the sequence-only Transformer performs substantially worse (0.08).

**Protein–nucleic acid and RNA-only complexes.** We evaluate our models on the protein–nucleic acid dataset from the AlphaFold-3 paper. Of the 199 structures, 172 pass our data pipeline. We also consider a subset of 27 RNA-only structures. For RNA-only complexes, we assess folding quality using lDDT, while for protein–nucleic acid complexes we evaluate interface accuracy using Interface Contact Similarity (ICS) and Interface Patch Similarity (IPS). We compare large Pairformer, Pairmixer, and Transformer models trained for the same number of iterations in Table 1d and Table 1c. On protein–nucleic acid complexes, *Pairmixer* performs comparably to Pairformer, with the Transformer lagging behind. For RNA-only structures, *Pairmixer* again matches Pairformer, while the Transformer performs better, likely due to limited RNA structural training data. Notably, Pairmixer achieves performance comparable to Pairformer despite removing sequence attention from the trunk.

### C.4 DETAILS OF PAIRMIXER APPLIED TO PROTEIN DESIGN

BindCraft (Pacesa et al., 2025), BoltzDesign (Cho et al., 2025), and hallucination-based protein design methods (Frank et al., 2024; Wicky et al., 2022; Jendrusch et al., 2025; Goverde et al., 2023; Bryant & Elofsson, 2022; Anishchenko et al., 2021) have recently demonstrated that structure predictors can be repurposed as differentiable scoring functions for sequence optimization. The input sequence is treated as a set of learnable parameters and is updated by backpropagating through a structure predictor, thereby jointly refining sequence and structure toward favorable interactions with the target protein or small molecule. While powerful, these methods have practical limitations: memory demands are high and sequence generation is slow, requiring hundreds of runs of the structure predictor per design. This inefficiency makes the approach prohibitively expensive, particularly for larger systems.

To address these challenges, we introduce BindFast, a scalable and efficient framework for binder design which replaces BoltzDesign's Pairformer backbone with *Pairmixer*. BindFast substantially reduces the runtime and memory footprint of binder generation and aim to accelerate the discovery of high-quality binders, particularly for large targets.

In Table 2, we benchmark the runtime performance of BindFast against BoltzDesign for generating 110-residue binders across a range of target proteins with biotechnological relevance, using an A100 GPU with 80 GB memory. BoltzDesign failed with out-of-memory (OOM) errors on targets larger than 500 residues, whereas BindFast extended this limit to 650 residues, a 30% improvement in target size. For protein targets where both models executed without memory overflow, BindFast achieves speedups of 2x to 2.6x at total sequence lengths ranging from 140 to 607, respectively. Qualitative comparisons in Figure 13 further indicate that BindFast produces designs comparable to those of BoltzDesign, underscoring its potential for faster in-silico iteration and enabling the design of binders against larger, more biologically relevant targets.

Table 6: **Pairformer Ablation.** We remove each module in the Pairformer one at a time.

| Ablation | GPU days | lDDT | DOCKQ$_{>0.23}$ | lDDT$_{PLI}$ | RMSD$_{<2}$ |
|---|---|---|---|---|---|
| - | 82 | 0.74 | 0.57 | 0.52 | 0.50 |
| No Seq Update | 80 | 0.73 | 0.57 | 0.54 | 0.49 |
| No Tri Att | 66 | 0.70 | 0.55 | 0.50 | 0.48 |
| No Tri Mul | 71 | 0.70 | 0.53 | 0.49 | 0.46 |

# D    ADDITIONAL ANALYSIS

## D.1    PAIRFORMER ABLATIONS

We performed ablation experiments on a small 12-layer Pairformer model to isolate the contributions of triangle multiplication, triangle attention, and sequence updates in Table 6. The results show that, under a short training schedule of 60 epochs (3M samples), both triangle multiplication and triangle attention are essential for performance, while sequence updates have minimal impact. Notably, *Pairmixer* recovers performance with additional training.

## D.2    ADDITIONAL ABLATIONS

**Triangle Multiplication vs. Feed-Forward Network.** We aim to understand how the performance is affected by the triangle multiplication and pair feed-forward networks, the two core ingredients of the *Pairmixer* architecture. In Table 7a and Table 7b, we vary the hidden dimensions of these components to evaluate model's sensitivity. For the FFN, we change the hidden dimension that the model expands to. For triangle multiplication, we instead project the features into higher- or lower-dimensional spaces before the multiplication and then project them back to the input dimension. We find that decreasing the FFN hidden dimension does not change performance much, while doubling the FFN dimension increases the mean lDDT from 0.71 to 0.74. We see a similar trend with triangle multiplication dimensions – doubling the hidden dimension improves the mean lDDT from 0.71 to 0.73, while reducing the dimensionality does not change lDDT.

**Other mixing methods.** Triangle multiplication mixes features within the $z \in \mathbb{R}^{L \times L \times D}$ pair representation. In Table 7c, we replace this operation with alternative, simpler mixing functions. First, we ablate the outgoing triangle multiplications, retaining only the incoming variant. Second, we introduce an FFT mixer that applies the discrete Fourier transform along rows and columns, following FNet (Lee-Thorp et al., 2021). Finally, we test a pooling mixer that averages representations across each row (and column) and adds the result back to all positions along the corresponding axis.

We find that these simplified approaches are insufficient and underperform compared to vanilla triangle multiplication. For instance, the FFT mixer likely fails because it mixes features solely based on sequence position, ignoring discontinuities introduced by multiple chains.

Table 7: *Pairmixer* **ablations experiments.** Default settings are marked in grey. See Section D.2 for details. $D_p$: number of pairmixer layers. $D_d$: number of diffusion transformer layers.

(a) **FFN Hidden Dimension**

| dim | lDDT | DOCKQ$_{>0.49}$ | lDDT$_{PLI}$ | RMSD$_{<1}$ |
|---|---|---|---|---|
| 256 | 0.71 | 0.38 | 0.50 | 0.34 |
| 512 | 0.71 | **0.42** | 0.50 | 0.33 |
| 1024 | **0.74** | 0.40 | **0.53** | **0.35** |

(b) **Triangle Mul Dimension**

| dim | lDDT | DOCKQ$_{>0.49}$ | lDDT$_{PLI}$ | RMSD$_{<1}$ |
|---|---|---|---|---|
| 64 | 0.71 | 0.41 | 0.50 | 0.34 |
| 128 | 0.71 | **0.42** | 0.50 | 0.33 |
| 256 | **0.73** | **0.42** | **0.52** | **0.37** |

(c) **Mixing Method**

| mixer | lDDT | DOCKQ$_{>0.49}$ | lDDT$_{PLI}$ | RMSD$_{<1}$ |
|---|---|---|---|---|
| FFT | 0.66 | 0.34 | 0.45 | 0.27 |
| AvgPool | 0.69 | 0.35 | 0.48 | 0.31 |
| TriMul-rows | 0.70 | 0.35 | 0.49 | 0.32 |
| TriMul-both | **0.71** | **0.42** | **0.50** | **0.33** |

(d) **Diffusion Transformer Depth**

| $D_p$ | $D_d$ | lDDT | DOCKQ$_{>0.49}$ | lDDT$_{PLI}$ | RMSD$_{<1}$ |
|---|---|---|---|---|---|
| 12 | 12 | 0.73 | 0.43 | 0.52 | 0.34 |
| 24 | 24 | **0.75** | **0.45** | **0.54** | **0.40** |

**Diffusion Module.** The diffusion module takes the latent representations as input and decodes the 3-dimensional protein structure using a 24-layer transformer. In Table 7, we evaluate how sensitive the Pairformer and *Pairmixer* architectures are to the size of the diffusion module.

### D.3    TRIANGLE MULTIPLICATION APPLIED TO THE MATCH3 TASK

To separate the effect of quadratic pair representations from the cubic cost of triangle operations, we evaluate these components on the Match3 task, a benchmark for learning 3-way interactions (Sanford et al., 2023; Kozachinskiy et al., 2025). Match3 gives the model a sequence $\mathbf{x} \in [M]^N$ and asks whether any triple of distinct elements sums to zero mod $M$. We use $N = 16$, $M = 64$, a hidden dimension of 8, comparable parameter counts (900–1100), and standard embedding, projection, and max-pooling layers, training on balanced datasets.

We compare three architectures that separate representational and computational factors: standard self-attention (linear representations, quadratic compute), third-order self-attention (linear representations, cubic compute (Roy et al., 2025)), and triangle multiplication (quadratic pair representations, cubic compute). This setup isolates whether performance gains stem from the richer pair representation or from cubic-order computation.

Across data regimes, all architectures struggle under extreme data scarcity. However, as data and depth increase, standard self-attention consistently lags behind both cubic-compute methods (see Figure 14). Notably, triangle multiplication on quadratic pair representations outperforms both variants of Transformer-style attention in shallow settings, demonstrating a particular advantage in capturing nonlocal 3-token interactions even when computational budgets are matched.

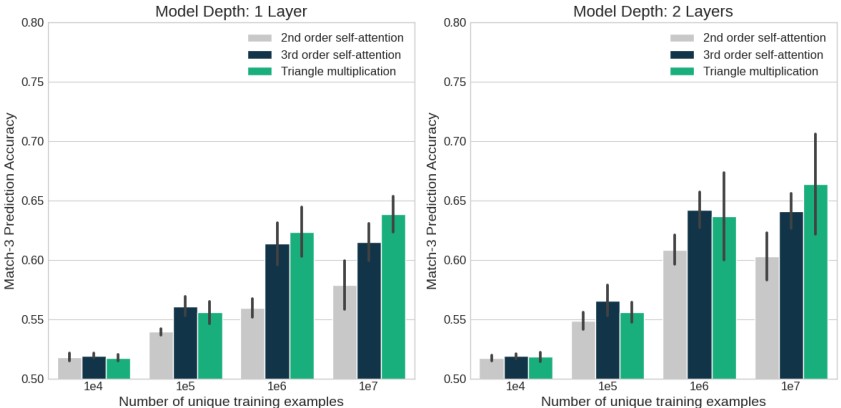

Figure 14: **Comparison between architecture variants on Match3.** We report classification accuracy on Match3 task as a function of training data size and model depth.

### D.4 DISSECTING PAIRMIXER PERFORMANCE ON THE RCSB TEST SET

To gain deeper insight into these results, we analyze Pairmixer, Pairformer, and Transformer performance across the RCSB test set through two complementary perspectives.

We first examine the correlation between Pairformer and Pairmixer performances. In Figure 15, each point represents a single structure, with coordinates indicating the respective model's performance. The strong correlation between the two models, with minimal outliers, suggests the two architectures share similar failure and success modes. Furthermore, Pairmixer outperforms Pairformer on lDDT in 44.9% of the test cases, demonstrating near-equivalent predictive capability despite the architectural simplification.

Next, we investigate whether Pairmixer's competitive performance is confined to favorable conditions, specifically, short sequences or proteins with abundant homologous sequences. We partition the RCSB test set by sequence length and MSA depth, then evaluate all three models across these stratified subsets in Figure 16. As expected, all models achieve higher accuracy on shorter proteins and those with richer MSA information, while accuracy degrades for longer sequences and sparser alignments. Critically, these performance trends remain consistent across architectures, with Pairmixer maintaining parity with Pairformer across all difficulty regimes.

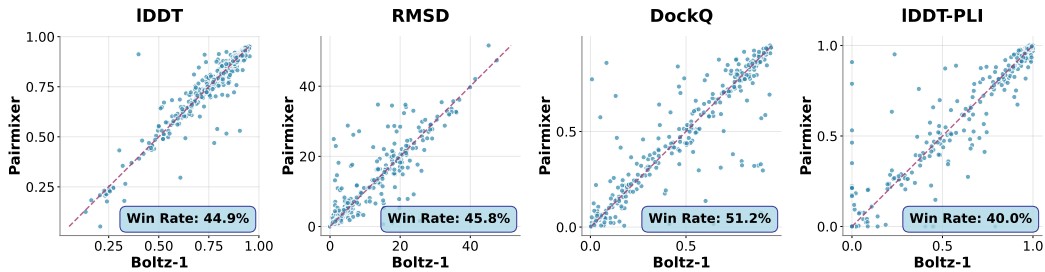

Figure 15: **Head-to-head comparison between *Pairmixer* and the Pairformer backbone.** The win rate shows how often the *Pairmixer* architecture achieves a better score than the Transformer architecture. In 89% of cases, the two models' lDDT scores differ by less than 5 points.

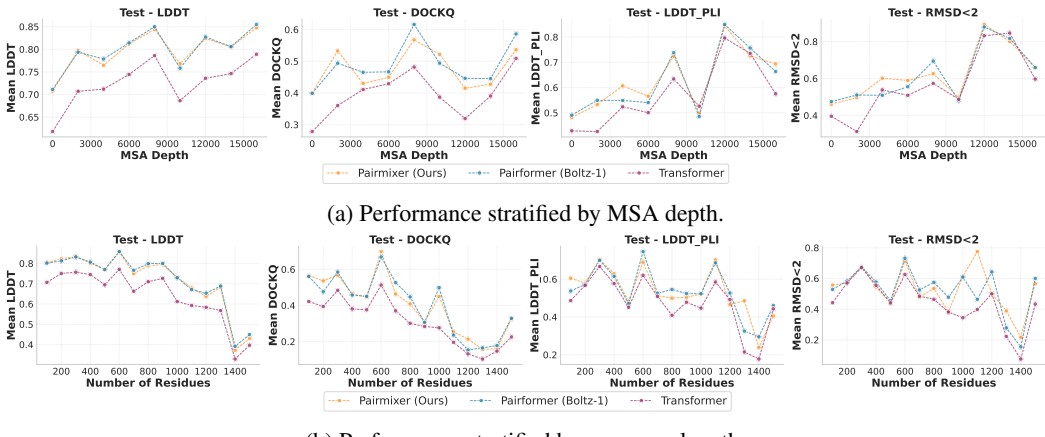

(a) Performance stratified by MSA depth.

(b) Performance stratified by sequence length.

Figure 16: **Performance across different data difficulty metrics.** Pairmixer maintains comparable performance to Pairformer across all difficulty levels.

## D.5 DISTOGRAM PERFORMANCE

A potential confounding factor in evaluating Pairmixer is that the diffusion module may correct errors in lower-quality backbone outputs, producing high-quality structures independently of the backbone. To isolate the backbone's contribution, we evaluate distogram predictions on the RCSB test set. The distogram head predicts a 64-bin discretized distance matrix directly from backbone features $z^{backbone}$, and accuracy is measured both across the full system and between chains in Table 8a. Following standard contact prediction evaluation (Moult et al., 2014), we also report precision at $L$ and $L/5$ in Table 8b. Across all metrics, *Pairmixer* performs comparably to Pairformer, suggesting that the Pairmixer with triangle multiplication and feed-forward networks produce equally expressive backbone features.

Table 8: **Distogram Prediction Performance.**

(a) Global and Inter-Chain Accuracy

| Method | Global | | Inter-Chain | |
|---|---|---|---|---|
| | Top-1 Acc | Top-5 Acc | Top-1 Acc | Top-5 Acc |
| Pairformer (Boltz-1) | 0.74 | 0.89 | 0.67 | 0.73 |
| Pairmixer (Ours) | 0.73 | 0.88 | 0.67 | 0.73 |
| Transformer | 0.72 | 0.86 | 0.67 | 0.72 |

(b) Contact Prediction

| Method | Short | | Medium | | Long | |
|---|---|---|---|---|---|---|
| | P@L | P@L/5 | P@L | P@L/5 | P@L | P@L/5 |
| Pairformer (Boltz-1) | 0.72 | 0.75 | 0.72 | 0.76 | 0.73 | 0.81 |
| Pairmixer (Ours) | 0.72 | 0.75 | 0.72 | 0.76 | 0.73 | 0.80 |
| Transformer | 0.69 | 0.72 | 0.69 | 0.74 | 0.70 | 0.79 |

## D.6 SENSITIVITY ANALYSIS TO NUMBER OF RECYCLING STEPS

To ensure Pairmixer's performance is not solely due to greater benefits from recycling, we evaluate Pairformer, Pairmixer, and Transformer with 0, 1, 3, and 10 recycling steps in Table 9. All models use the large setting (48 layers) and are trained for the same number of iterations. Pairformer and Pairmixer achieve similar results across different numbers of recycles, suggesting that Pairmixer's comparable performance is not simply a result of additional recycling.

Table 9: **Impact of Recycling Steps**. Performance increases but quickly saturates for all architectures.

| Architecture | Recycles | lDDT | $DOCKQ_{>0.23}$ | $lDDT_{PLI}$ | $RMSD_{<2}$ |
|---|---|---|---|---|---|
| Pairformer (Boltz-1) | 0 | 0.75 | 0.59 | 0.55 | 0.53 |
| Pairformer (Boltz-1) | 1 | 0.77 | 0.59 | 0.58 | 0.56 |
| Pairformer (Boltz-1) | 3 | 0.78 | 0.62 | 0.57 | 0.55 |
| Pairformer (Boltz-1) | 10 | 0.78 | 0.64 | 0.57 | 0.54 |
| Pairmixer (Ours) | 0 | 0.74 | 0.59 | 0.54 | 0.52 |
| Pairmixer (Ours) | 1 | 0.76 | 0.61 | 0.56 | 0.56 |
| Pairmixer (Ours) | 3 | 0.77 | 0.61 | 0.56 | 0.54 |
| Pairmixer (Ours) | 10 | 0.78 | 0.63 | 0.57 | 0.55 |
| Transformer | 0 | 0.62 | 0.40 | 0.48 | 0.47 |
| Transformer | 1 | 0.67 | 0.49 | 0.51 | 0.48 |
| Transformer | 3 | 0.70 | 0.52 | 0.52 | 0.49 |
| Transformer | 10 | 0.70 | 0.53 | 0.51 | 0.48 |

# E    MODEL HYPERPARAMETERS

Table 10 includes a thorough list of the hyperparameters used for our experiments. This table additionally includes the training FLOPs for all model architectures and sizes.

Table 10: **Model Hyperparameters.** Dashes (-) indicate same value as the previous column. The large model is trained with smaller crops and mixed data, then with larger crops and PDB-only data.

| Hyperparameter | Small | Medium | Large Stage 1 | Large Stage 2 |
|---|---|---|---|---|
| *Model Architecture* | | | | |
| Number of Backbone Layers | 12 | 24 | 48 | 48 |
| Number of MSA Layers | 4 | - | - | - |
| Token representation dim ($C_s$) | 384 | - | - | - |
| Pair representation dim ($C_z$) | 128 | - | - | - |
| Backbone dropout | 0.25 | - | - | - |
| MSA Module dropout | 0.15 | - | - | - |
| Number of Diffusion Layers | 6 | 24 | 24 | 24 |
| Atom representation dim | 128 | - | - | - |
| Atom pair representation dim | 16 | - | - | - |
| *Training* | | | | |
| Optimizer | Adam | - | - | - |
| Maximum learning rate | $1.8 \times 10^{-3}$ | - | - | - |
| Diffusion multiplicity | 16 | - | - | - |
| Recycling | 0,1,2,3 | - | - | - |
| Epochs | 68 | 68 | 68 | 20 |
| Training Samples | 6.8M | 6.8M | 6.8M | 2M |
| *Data Processing* | | | | |
| Data source | PDB + OpenFold | - | - | PDB |
| Maximum tokens | 384 | 384 | 384 | 512 |
| Maximum atoms | 3,456 | 3,456 | 3,456 | 4,608 |
| Maximum MSA sequences | 2,048 | - | - | - |
| Samples per epoch | 100,000 | - | - | - |
| Total Batch size | 128 | - | - | - |
| *Inference* | | | | |
| Number of sampling steps | 200 | - | - | - |
| Maximum MSA Sequences | 4096 | - | - | - |
| Recycling | 10 | - | - | - |
| Diffusion samples | 5 | - | - | - |
| *Training Infrastructure* | | | | |
| GPU Type | H200 | - | - | - |
| Number of GPUs | 32 | 32 | 32 | 64 |
| *Total Training FLOPs* | | | | |
| Boltz-1 (Pairformer) | 8.306e+19 | 1.467e+20 | 2.707e+20 | 1.572e+20 |
| *Pairmixer* | 4.817e+19 | 8.557e+19 | 1.572e+20 | 8.716e+19 |
| Transformer | 5.784e+18 | 7.888e+18 | 8.941e+18 | 4.205e+18 |
| *Inference FLOPs* | | | | |
| Boltz1 (Pairformer) | 9.100e+15 | 1.595e+16 | 2.964e+16 | - |
| *Pairmixer* | 4.474e+15 | 7.849e+15 | 1.460e+16 | - |
| Transformer | 4.137e+14 | 4.975e+14 | 6.652e+14 | - |

