# OpenReview forum: "Triangle Multiplication is All You Need for Biomolecular Structure Representations"
_ICLR.cc/2026/Conference — ICLR 2026 Poster_

### Official Review · Reviewer_J8sg · 2025-10-16

**Soundness:** 3
**Presentation:** 4
**Contribution:** 2
**Rating:** 4
**Confidence:** 4

**Summary:**

This work applies triangle multiplication in the large scale of current biomolecular structure prediction. Such application scenario has not been investigated with models using triangle multiplication. The authors show that, for the Pairformer backbone, when maintaining triangle multiplication, and omitting triangle attention, the training and inference efficiency can be improved.

**Strengths:**

1. The paper is well-written, clearly depicting the methodology;
2. The training cost and inference speed are both improved;
3. The large-scale biomolecular structure prediction scenario is of great practical importance;
4. The experiments are sufficient.

**Weaknesses:**

1. This work lacks of methodological contribution. The authors omit other modules and maintaining only the triangle multiplication. This modification is considered very trivial and not that novel. The importance of triangle multiplication has already been investigated by several proceeders, as the authors themselves claimed in the paper. The author is the first one deleting triangle attention for PairFormer.  The main contribution is testing if we can achieve better performance-efficiency trade-off using only triangle multiplication.
2. The efficiency improvement is not that satisfying. To be fair, it is adequate when the methodological contribution is enough. But since the contribution is minor, I would expect giant efficiency leap to complement the limited contribution.

**Questions:**

Whether there is any non-trivial part for deleting triangle attention and other modules? This could be a potential methodological contribution if there is any.

Suggestion: If my evaluation remains consistent after I read the authors' response, I would suggest that the author considers submitting this work to an application-oriented venue, e.g., Nat. Comm., Nat. Comp. Sci., Sci. Adv. The value of this work is mostly about the application results but without meaningful methodological insights. After giving stronger application evaluations, packing as a tool box or executable empirical platform, this work is more suitable to those top application-oriented venues. The methodological novelty, in my opinion, is enough for them.

I am also open to change my mind if the author can prove the methodological value that I failed to see.

---

> ### Author Response · Authors · 2025-11-22
> **Thanks for the review.**
>
> We thank the reviewer for recognizing the practical importance and results of the work.
> We respectfully clarify the methodological contribution of Pairmixer. While prior works (Genie2, Proteina) show that triangle multiplication can improve sequence transformers, and MiniFold suggests it can match ESMFold, **these findings do not establish that triangle multiplication alone can serve as the core operator in a large-scale Pairformer architecture**. Indeed, all current state-of-the-art biomolecular structure predictors, including AlphaFold3, Boltz-2, Chai, and Protenix, continue to rely on triangle attention. Demonstrating that triangle multiplication alone can replace triangle attention at scale, in practice, and without loss of accuracy is a non-trivial conceptual contribution.
>
> The resulting model is faster, but this efficiency is a consequence of a new scientific insight, not the goal itself.
>
> Furthermore, we provide **methodological insights** into why triangle multiplication is effective for biomolecular structure prediction. Our analysis shows that
> 1. Pairmixer surpasses transformers specifically in modeling pairwise interactions. (See Section 6)
> 2. Triangle multiplication excels at capturing sparse, long-range geometric interactions. (See Supplement Section F)
> 3. The sequence track in Pairformer does not meaningfully improve the performance of the model. (See Supplement Section H)
>
> These analyses offer concrete insights that will inform the design of the next generation of structure-prediction architectures.
>
> We also emphasize that our evaluation is broad and rigorous. We additionally evaluate Pairmixer on the AF3 antibody-antigen test set, AF3 nucleic acid test set, and PoseBusters, consistently observing parity with state-of-the-art (full results available in supplement Section G). We further validate the effectiveness of Pairmixer across various experimental settings (full results available in supplement Section I).
>
> To summarize, our results demonstrate that a trunk composed solely of triangle multiplication + feed-forward networks achieves state-of-the-art accuracy and our analyses explain why this simplified architecture works. In view of these contributions, we believe the work clearly satisfies the ICLR criterion of “contributing new knowledge and sufficient value to the community” (ICLR Reviewer Guide https://iclr.cc/Conferences/2026/ReviewerGuide). We hope the reviewer will find our clarification convincing.

---

> ### Comment · Reviewer_J8sg · 2025-11-28
>
> Thank you for the response. Section F helps in discovering the mechanism behind, making this paper more than a technical report. I am willing to raise my score from 4 to 6. Unfortunately, the deadline of review revision has passed, so I just express my opinion here for AC's consideration.

---

### Official Review · Reviewer_GgDp · 2025-10-23

**Soundness:** 3
**Presentation:** 4
**Contribution:** 2
**Rating:** 2
**Confidence:** 4

**Summary:**

This paper addresses the computational efficiency issue in AlphaFold and argues that the triangle attention operation contributes little to the final predictive performance while introducing significant computational overhead. Based on this observation, the authors propose the PairMixer block, a simplified variant of the Pairformer block used in AlphaFold3, in which operations with minimal performance benefits are removed. Experimental results on the RCSB test set demonstrate that PairMixer achieves comparable performance to the original Pairformer while offering up to a 4x improvement in inference speed.

**Strengths:**

1. This paper focuses on an important research problem, i.e., to accelerate AlphaFold and make it more lightweight, which is critital for down-stream applications like virtual screening;
2. This paper is well-written and clearly-structured. The figures effectively support the understanding of the proposed method, and the authors provide sufficient background and preliminaries to contextualize their work.

**Weaknesses:**

1. The contribution of this paper appears limited. The proposed method can be viewed primarily as an engineering optimization of the original Pairformer, without introducing substantial new insights. Without deeper analysis or justification of the design choices, the current contribution may not meet the novelty threshold typically expected for a venue such as ICLR. Furthermore, the finding that the triangle attention module contributes minimally to performance is not particularly surprising; this has been informally noted by several researchers through ablation studies, even though such observations have not been formally published.

2. The experimental evaluation of PairMixer is insufficient. The results are reported only on the RCSB dataset (533 structures), which limits the generalizability of the conclusions. It is recommended that the authors adopt a broader evaluation protocol, such as the one used in Boltz-2, to strengthen the empirical validation of their method.

3. The use of the phrase “all you need” in the title, while common in machine learning literature, is not appropriate in this context. The PairMixer/Pairformer serves only as the model trunk within a structure prediction framework, whereas the diffusion module plays an equally important role. Therefore, the current title may overstate the scope and completeness of the proposed contribution.

**Questions:**

No further questions. Please see the “Weaknesses” section for details.

---

> ### Author Response · Authors · 2025-11-22
> **Thanks for the review. 1/2**
>
> We are glad that the reviewer finds our research problem important and that the background and preliminaries are sufficient.
>
> **Contribution is an “engineering optimization.”**
> We thank the reviewer for their thoughtful feedback. We respectfully disagree with the characterization of our contribution as “primarily an engineering optimization.” Our core contribution is scientific: we identify the minimum computational mechanisms required in the structure predictor’s trunk. More specifically, we show that the complex and computationally expensive triangle attention used in virtually all state-of-the-art biomolecular structure predictors is not conceptually essential. Instead, the trunk can be reduced to a far simpler operation (Pairmixer) without a meaningful loss in accuracy. This is a conceptual simplification that offers new insight into the fundamental roles of pairwise updates in structure prediction.
>
> Our analysis further demonstrates why this simplification works. We show that triangle multiplication is equally capable of producing geometrically consistent representations and even learns sparse, long-range interactions (Supp. Sec. F). Moreover, Pairmixer surpasses Transformers specifically on modeling pairwise interactions, providing an additional insight into why the trunk can operate effectively without a sequence update (Sec. 6).
>
> The resulting model is faster, but this efficiency is a consequence of a new scientific insight, not the goal itself.
>
> **Results are not surprising because they have informally noted by several researchers through ablation studies, even though such observations have not been formally published.**
> We appreciate the reviewer's comment that the minimal contribution of triangle attention may be 'informally noted' by researchers. However, as the reviewer acknowledged, no prior published work has formally demonstrated, rigorously benchmarked, and analyzed this finding on a SOTA-scale model. A key contribution of our paper is to take this 'informal observation' and provide the first concrete, empirical evidence to support it, moving it from community 'folklore' to a published, actionable scientific finding.
>
> Furthermore, our contribution is constructive. We do not simply ablate the modules and report a small change. We propose a novel, successful, and highly efficient replacement. Identifying a redundancy is one step; designing a superior alternative that leverages this insight is a significant contribution in its own right. We are confident that this insight, combined with our novel Pairmixer module, will be highly valuable for the community in building more efficient and scalable structure predictors, and we believe this contribution clearly meets the novelty threshold for ICLR.
>
> **The use of “All You Need” in the title is not appropriate.**
> Thank you for your suggestion. We understand the concern. We would argue that in our case, it is as appropriate as in the original transformer paper. Both papers use the phrase “All You Need” to highlight a simple core component of their architecture. Yes, the transformer has other parts (MLP, residuals, normalizations, …). However, attention was its core building block. This is the same for our architecture. We thus believe “All You Need” as used and introduced by “Attention is All You Need” is appropriate in this context.

---

> > ### Author Response · Authors · 2025-11-22
> > **Thanks for the review. 2/2**
> >
> > **Insufficient experimental evaluation.**
> > To further strengthen our evaluation, we have added new datasets covering antibodies, protein-RNA complexes, RNA-only structures, and protein-ligand structure prediction. The results suggest that Pairmixer performs on par with Pairformer-based structure predictors. These results can be found in the updated paper in Supplement Section G.
> >
> > Posebusters (298 structures)
> > | Method | RMSD < 2         | RMSD < 1         | P-L LDDT         |
> > |--------|------------------|------------------|------------------|
> > | Pairmixer (Ours)     | 0.672 ± 0.028    | 0.449 ± 0.029    | 0.733 ± 0.014    |
> > | Pairformer (Boltz-1)  | 0.682 ± 0.027    | 0.456 ± 0.029    | 0.737 ± 0.015    |
> >
> > Antibodies (71 structures)
> > | Method                        |   DockQ_>0.23 |
> > |:------------------------------|--------------:|
> > | Pairformer (Boltz-1)          |          0.23 |
> > | Pairmixer (Ours)              |          0.23 |
> > | Transformer                   |          0.08 |
> >
> > Protein-RNA complexes (172 structures)
> > | Method                 |   ICS |   IPS |
> > |:-----------------------|------:|------:|
> > | Pairformer (Boltz-1)   |  0.50 |  0.65 |
> > | Pairmixer (Ours)       |  0.51 |  0.66 |
> > | Transformer            |  0.48 |  0.64 |
> >
> > RNA only (27 structures)
> > | Method                   |   LDDT |
> > |:-------------------------|-------:|
> > | Pairformer (Boltz-1)     |   0.58 |
> > | Pairmixer (Ours)         |   0.59 |
> > | Transformer              |   0.61 |
> >
> > To further strengthen the empirical validation of the method and extend the generalization of our conclusions, we performed extensive evaluations varying the number of recycles, measuring distogram performance, and analyzing performance as a function of sequence length and MSA depth. Across these settings, Pairmixer performs comparably to Pairformer, suggesting that triangle multiplication captures representations of similar quality as triangle attention. Full results are made available in Supplement Section I in the revised paper.

---

> > > ### Comment · Reviewer_GgDp · 2025-11-26
> > > **Post Rebuttal**
> > >
> > > Thanks the authors for the responses to my questions, and the additional experimental results. Some of my concerns are addressed. However, based on the technical contribution and performance, I still do not believe this paper meets the threshold for ICLR. I will raise my rating from 2 to 4.

---

### Official Review · Reviewer_r3YY · 2025-10-30

**Soundness:** 3
**Presentation:** 3
**Contribution:** 3
**Rating:** 6
**Confidence:** 3

**Summary:**

The paper proposes Pairmixer, a modified AlphaFold3-style backbone that (a) deletes triangle attention and sequence updates from the Pairformer backbone, (b) keeps only triangle multiplication + pairwise FFNs to update the pair representation, and (c) leaves the single-sequence representation unchanged and feeds it directly to the downstream diffusion module. The experiments are shown with maintaining folding / docking accuracy while significantly improving speed (up to 4× faster inference on 2048-token sequences and 34% less training compute), and unlock downstream design workflows that previously ran out of memory.

**Strengths:**

Reviewer appreciates the following contributions:

- **impactful and practical**: the paper addresses a real bottleneck in AlphaFold-style models by removing triangle attention for faster, more scalable inference and training. Furthermore, it enables long-sequence and large-complex modeling that was previously infeasible due to memory or computational limits.

- **Empirical validation**: Demonstrates near-identical accuracy to AlphaFold3-class baselines across folding, docking, and binder design tasks, with up to 4× speedup and lower memory use. Experiments include comparisons on Boltz-1, Transformer, and ablations, FLOPs analysis, and realistic large-scale design benchmarks.

- **Simple but be efficient**: the paper shows that triangle multiplication alone suffices for capturing higher-order geometric consistency, providing a clearer understanding of what inductive biases matter. In particular, the final architecture will be the form of *Pairmixer = triangle multiplication + FFN over z, recycle it N times*.

- **presentation**: the paper is well-written and easy to follow.

**Weaknesses:**

- **Method Novelty**: Novelty somewhat incremental: Prior works (e.g., Genie2, MiniFold) already suggested triangle multiplication is key; this paper mainly extends the idea to AF3 scale rather than introducing it conceptually. This requires further analysis to highlight key differences between Pairmixer versus prior works, for e.g., what should we do to adapt for the protein structure design task?

- **Limited generalization tests:**
While the paper benchmarks extensively on protein–protein and protein–ligand systems, all evaluations remain within domains similar to the training distribution of Boltz-1 (PDB-scale protein complexes). The work does not assess generalization to more diverse biomolecular systems, such as RNA–protein assemblies, RNA-only structures, metalloproteins, or highly flexible/transient complexes. These categories often require different geometric reasoning and long-range constraints, where triangle attention might still provide advantages. Without such tests, it’s quite unclear whether the proposed architecture truly generalizes beyond well-structured protein complexes, or if its performance degrades on systems with **unconventional topologies or more dynamic conformational behavior**.

**Questions:**

**Missing sequence update ablation:**
The paper removes sequence updates but doesn’t isolate their effect. It’s unclear how much of the performance change comes from dropping triangle attention versus removing sequence updates.

**Lack of discussion on limitations vs AlphaFold3:**
Can authors discuss where the simplified model may fail compared to full AlphaFold3 — for example, in modeling highly flexible regions, RNA–protein complexes, or subtle side-chain rearrangements requiring long-range attention?

---

> ### Author Response · Authors · 2025-11-22
> **Thanks for the review. 1/2**
>
> We are pleased that the reviewer finds our work both impactful and practical, and appreciates our empirical validation.
>
> **Novelty.**
> We thank the reviewer for placing our work in the context of prior art such as Genie2 and MiniFold. While these works demonstrate effectiveness on smaller models, techniques that work in smaller or specialized models do not always transfer successfully to deep, all-atom architectures. This challenge is underscored by the fact that nearly all state-of-the-art predictors developed after these prior works, including AlphaFold3, Boltz-2, Chai, and Protenix, still rely on triangle attention. Our paper’s contribution is to develop a simplified model that performs on par with state-of-the-art structure predictors.
>
>
> **Sequence update ablation.**
> We performed ablation experiments on a small Pairformer model to isolate the contributions of triangle multiplication, triangle attention, and sequence updates. The results show that, under a short training schedule of 60 epochs (3M samples), both triangle multiplication and triangle attention are essential for performance, while sequence updates have minimal impact. Notably, models without triangle attention recover performance with additional training. These results have been added to the paper in supplement Section H.
> | Size   | Ablation      |   GPU days |   lDDT |   DockQ>0.23 |   lDDT-PLI |   RMSD<2Å |
> |:-------|:--------------|-----------:|-------:|-------------:|-----------:|----------:|
> | Small  |               |      81.51 |   0.74 |         0.57 |       0.52 |      0.50 |
> | Small  | No Seq Update |      79.64 |   0.73 |         0.57 |       0.54 |      0.49 |
> | Small  | No Tri Att    |      65.94 |   0.70 |         0.55 |       0.50 |      0.48 |
> | Small  | No Tri Mul    |      70.95 |   0.70 |         0.53 |       0.49 |      0.46 |
>
> **Generalization to diverse biomolecular systems.**
> We thank the reviewer for highlighting the importance of evaluating generalization to diverse biomolecular systems. We evaluated Pairmixer on RNA-only structures and protein–RNA complexes in the AF3 nucleic acid evaluation dataset.
>
> On protein–RNA complexes, Pairmixer performs comparably to Pairformer, suggesting that triangle multiplication effectively supports geometric reasoning in these systems. On RNA-only structures, both Pairmixer and Pairformer perform below the Transformer baseline, likely due to the limited availability of RNA structural data. Notably, Pairmixer achieves performance on par with Pairformer despite lacking sequence attention in the trunk. We have incorporated these results into the paper in supplement Section G.
>
> RNA only (27 structures)
> | Method                   |   LDDT |
> |:-------------------------|-------:|
> | Pairformer (Boltz-1)     |   0.58 |
> | Pairmixer (Ours)         |   0.59 |
> | Transformer              |   0.61 |
>
> Protein-RNA complexes (172 structures)
> | Method                 |   ICS |   IPS |
> |:-----------------------|------:|------:|
> | Pairformer (Boltz-1)   |  0.50 |  0.65 |
> | Pairmixer (Ours)       |  0.51 |  0.66 |
> | Transformer            |  0.48 |  0.64 |

---

> > ### Author Response · Authors · 2025-11-22
> > **Thanks for the review. 2/2**
> >
> > **Limitations vs AlphaFold3’s Pairformer.**
> > We thank the reviewer for the suggestion to discuss limitations on flexible systems. On RNA-only structures, both triangle-based Pairformer and Pairmixer perform worse than a sequence-only Transformer, highlighting that the pair representation may be less advantageous in these more data limited regimes.
> >
> > To better understand where Pairmixer may fall short compared to full AlphaFold3, we performed extensive evaluations varying the number of recycles, measuring distogram performance, and analyzing performance as a function of sequence length and MSA depth. Across these settings, Pairmixer performs comparably to Pairformer, suggesting that triangle multiplication captures representations of similar quality as triangle attention. Full results are made available in Supplement I in the revised paper.
> >
> > Ablating number of recycles:
> > | Method | Recycles | LDDT | DockQ>0.23 | LDDT-PLI | RMSD < 2Å |
> > |---|---|---|---|---|---|
> > | Pairformer (Boltz-1) | 0 | 0.75 | 0.59 | 0.55 | 0.53 |
> > | Pairformer (Boltz-1) | 1 | 0.77 | 0.59 | 0.58 | 0.56 |
> > | Pairformer (Boltz-1) | 3 | 0.78 | 0.62 | 0.57 | 0.55 |
> > | Pairformer (Boltz-1) | 10 | 0.78 | 0.64 | 0.57 | 0.54 |
> > | Pairmixer (Ours) | 0 | 0.74 | 0.59 | 0.54 | 0.52 |
> > | Pairmixer (Ours) | 1 | 0.76 | 0.61 | 0.56 | 0.56 |
> > | Pairmixer (Ours) | 3 | 0.77 | 0.61 | 0.56 | 0.54 |
> > | Pairmixer (Ours) | 10 | 0.78 | 0.63 | 0.57 | 0.55 |
> > | Transformer | 0 | 0.62 | 0.40 | 0.48 | 0.47 |
> > | Transformer | 1 | 0.67 | 0.49 | 0.51 | 0.48 |
> > | Transformer | 3 | 0.70 | 0.52 | 0.52 | 0.49 |
> > | Transformer | 10 | 0.70 | 0.53 | 0.51 | 0.48 |
> >
> >
> > Distogram Accuracy Metrics:
> > | Method               | Top-1 Acc    | Top-5 Acc    | Inter-Chain Top-1 Acc   | Inter-Chain Top-5 Acc   |
> > |:---------------------|:-------------|:-------------|:------------------------|:------------------------|
> > | Pairformer (Boltz-1) | 0.74         | 0.89         | 0.67                    | 0.73                    |
> > | Pairmixer (Ours)     | 0.73 | 0.88 | 0.67           | 0.73     |
> > | Transformer          | 0.72    | 0.86 | 0.67            | 0.72            |
> >
> > Distogram Precision Metrics:
> > | Method               | Short P@L    | Short P@L/5   | Medium P@L   | Medium P@L/5   | Long P@L     | Long P@L/5   |
> > |:---------------------|:-------------|:--------------|:-------------|:---------------|:-------------|:-------------|
> > | Pairformer (Boltz-1) | 0.72         | 0.75          | 0.72         | 0.76           | 0.73         | 0.81         |
> > | Pairmixer (Ours)     | 0.72 | 0.75  | 0.72   | 0.76     | 0.73  | 0.80   |
> > | Transformer          | 0.69  | 0.72    | 0.69   | 0.74   | 0.70   | 0.79   |

---

> > > ### Comment · Reviewer_r3YY · 2025-11-25
> > > **Response to authors**
> > >
> > > Thanks to the author for providing additional experiments, I would keep my current positive evaluation.

---

### Official Review · Reviewer_qqDT · 2025-11-01

**Soundness:** 3
**Presentation:** 2
**Contribution:** 2
**Rating:** 4
**Confidence:** 2

**Summary:**

This paper introduces a new operator, Triangle Multiplication, as a core mechanism for relational reasoning and geometric representation learning. The method is presented as a lightweight yet expressive alternative to traditional self-attention, aiming to capture higher-order interactions among triplets of entities efficiently. The authors apply this operation to tasks, showing that it can achieve competitive or improved performance compared to transformer-style baselines.

The conceptual idea is creative and well-motivated; however, the presentation lacks sufficient technical clarity, the experimental scope is somewhat limited, and the empirical analysis does not fully demonstrate the operator’s claimed generality.

**Strengths:**

1.	Interesting conceptual direction: The idea of moving from pairwise attention to triangle-based relational modeling is novel and aligns with emerging research on geometric and higher-order attention.
2.	Simplicity of the operator:  The formulation is elegant and could potentially be a computationally efficient substitute for attention in specific contexts.
3.	Potential for extension: The proposed mechanism could inspire further work in 3D molecular or graph-structured domains, where triplet relations are natural.
4.	Readable overall motivation:  The high-level rationale and related work are generally well-written.

**Weaknesses:**

- Limited Comparative Breadth

The experiments benchmark against a few baselines but omit several directly relevant contemporary models, including:
	1. Higher-order attention variants (e.g., Tensor Attention, Relational Transformer)
	2. Geometric and 3D reasoning frameworks (e.g., SE(3)-Transformer, EGNN)
	3. Diffusion-based relational models and equivariant graph networks.

Without these comparisons, it is difficult to judge whether Triangle Multiplication provides a fundamentally better abstraction or merely a reparameterization of higher-order attention.

- Lack of Theoretical Clarity

The mathematical definition of “Triangle Multiplication” is presented at a high level but lacks rigorous derivation or clear connection to known tensor operations:
	1. The operator’s expressive power (what functions it can approximate) is not discussed.
	2. No complexity analysis is provided—readers cannot tell if it scales better than self-attention for large N.

- Weak Empirical Validation

While the experiments show some improvement, they remain qualitative and dataset-limited:
	1. The selected tasks are small-scale and do not reflect real-world complexity.
	2. No ablation studies are provided to isolate the contribution of the triangle operator vs. other components.

**Questions:**

see weakness

---

> ### Author Response · Authors · 2025-11-22
> **Thank you for the review.**
>
> We thank the reviewer for recognizing the elegance of the model.
>
> **Limited Comparative Breadth.**
> We agree that higher-order attention variants, geometric models, diffusion-based relational models and equivariant graph networks are highly relevant. Adapting these architectures into fully competitive biomolecular structure predictors requires substantial engineering and could constitute a paper-level contribution on its own, rather than ablations. We are not aware of these ideas being explored in the scope of biomolecular structure prediction and if there are specific baselines, we would love to compare with them.
>
> To ensure meaningful evaluation, we focused on strong baselines widely used in biomolecular structure prediction to directly measure the impact of triangle multiplication in this domain. We note that many of the suggested architectures are compatible with our framework, and we hope to explore such comparisons in future work.
>
> **Expressive Power.**
> While a formal mathematical characterization of triangle multiplication is challenging, our empirical studies demonstrate that any function that can be expressed by triangle attention can also be represented using triangle multiplication. Moreover, we demonstrate that triangle multiplication can effectively approximate sparse functions despite its inherently dense computation: zeroing out 50–75% of the terms does not degrade downstream performance. Further details can be found in Supplement Section F of the updated paper.
>
> **Complexity Analysis.**
> Triangle multiplication, like triangle attention, scales cubically with sequence length (a detailed FLOPs analysis is available in supplement Section B). However, its GPU implementation uses only highly efficient matrix multiplications, which are parallelizable across all 576 tensor cores on an H100 GPU and are substantially faster in practice.
>
> **Real-world complexity of experiments.**
> We note that the tasks we consider are highly relevant to real-world biomolecular research, including drug discovery and structural biology. To further strengthen our evaluation, we have added new datasets with real-world relevance covering antibodies, protein-RNA complexes, RNA-only structures, and protein-ligand structure prediction. These results can also be found in the supplement  Section G.
>
> Antibodies (71 structures)
> | Method                        |   DockQ_>0.23 |
> |:------------------------------|--------------:|
> | Pairformer (Boltz-1)          |          0.23 |
> | Pairmixer (Ours)              |          0.23 |
> | Transformer                   |          0.08 |
>
> Protein-RNA complexes (172 structures)
> | Method                 |   ICS |   IPS |
> |:-----------------------|------:|------:|
> | Pairformer (Boltz-1)   |  0.50 |  0.65 |
> | Pairmixer (Ours)       |  0.51 |  0.66 |
> | Transformer            |  0.48 |  0.64 |
>
> RNA only (27 structures)
> | Method                   |   LDDT |
> |:-------------------------|-------:|
> | Pairformer (Boltz-1)     |   0.58 |
> | Pairmixer (Ours)         |   0.59 |
> | Transformer              |   0.61 |
>
> Posebusters (298 structures)
> | Method | RMSD < 2         | RMSD < 1         | P-L LDDT         |
> |--------|------------------|------------------|------------------|
> | Pairmixer (Ours)     | 0.672 ± 0.028    | 0.449 ± 0.029    | 0.733 ± 0.014    |
> | Pairformer (Boltz-1)  | 0.682 ± 0.027    | 0.456 ± 0.029    | 0.737 ± 0.015    |
>
> **Ablation studies.**
> We performed ablation experiments on a small Pairformer model to isolate the contributions of triangle multiplication, triangle attention, and sequence updates. The results show that, under a short training schedule of 60 epochs (3M samples), both triangle multiplication and triangle attention are essential for performance, while sequence updates have minimal impact. Notably, models without triangle attention recover performance with additional training. We added this to our paper in supplement Section H.
> | Size   | Ablation      |   GPU days |   lDDT |   DockQ>0.23 |   lDDT-PLI |   RMSD<2Å |
> |:-------|:--------------|-----------:|-------:|-------------:|-----------:|----------:|
> | Small  |               |      81.51 |   0.74 |         0.57 |       0.52 |      0.50 |
> | Small  | No Seq Update |      79.64 |   0.73 |         0.57 |       0.54 |      0.49 |
> | Small  | No Tri Att    |      65.94 |   0.70 |         0.55 |       0.50 |      0.48 |
> | Small  | No Tri Mul    |      70.95 |   0.70 |         0.53 |       0.49 |      0.46 |

---

> > ### Comment · Reviewer_qqDT · 2025-11-27
> > **Response**
> >
> > Thanks the authors. The additional experimental results seems to address most of my concerns. I will raise my rating from 4 to 6.

---

### Author Response · Authors · 2025-11-22
**Thanks to all the reviewers.**

We thank all the reviewers for their thoughtful and constructive feedback. We are encouraged that reviewers found the work practical and impactful (r3YY, GgDp, J8sg), appreciated its empirical strength (r3YY, J8sg), and valued the simplicity, clarity, and computational efficiency of the architecture (qqDT, r3YY, J8sg).

The primary concerns centered on (1) methodological novelty, given prior observations about the usefulness of triangle multiplication, and (2) requests for additional evaluations, ablations, and discussion of limitations. We summarize our responses here.

Our paper’s central contribution is not to reintroduce triangle multiplication, but to propose a new architecture for biomolecular structure predictions  composed of a Pairmixer trunk (with solely of triangle multiplication and feed-forward networks) and diffusion head. The new architecture includes new connections between the architecture components and **matches state-of-the-art structure predictors** while yielding computational gains. Prior works such as MiniFold and Genie2 explored triangle multiplications in smaller or different settings, but it was unclear whether those findings would translate to large-scale, all-atom biomolecular structure prediction. After all, current state-of-the-art predictors (AlphaFold3, Boltz-2, Chai, Protenix) still rely on triangle attention.

We acknowledge reviewer GgDp’s observation that “the finding… [has] been informally noted by several researchers through ablation studies, even though such observations have not been formally published.” Our work takes this informal understanding and establishes it as a **concrete, reproducible result at biomolecular complex scale**. We provide **rigorous evaluations across diverse domains**, including proteins, antibodies, RNA, protein–nucleic acid assemblies, and protein–ligand complexes, demonstrating that triangle multiplication can effectively replace triangle attention in each setting. In addition, we offer **methodological insights and detailed analyses** that clarify *why* this simplified architecture works, helping to elucidate the inductive biases underlying large-scale structure prediction.

In summary, we have substantially strengthened the paper in addressing all of the reviewer feedback:

✓ Added comprehensive evaluations on real-world and diverse benchmarks, including protein–ligand, protein–nucleic acid, RNA, and antibody datasets, to demonstrate the generality of our approach (Supplement Section G). [Reviewers qqDT/r3YY/GgDp/J8sg]

✓ Analyzed the expressive power of triangle multiplication to clarify why this simplified architecture performs so well (Supplement Section F). [Reviewers qqDT/J8sg]

✓ Included ablation studies on Pairformer modules to understand each of the model’s components (Supplement Section H). [Reviewers qqDT/r3YY]

✓ Evaluated performance across a range of experimental settings to identify potential limitations and failure modes (Supplement Section I). [Reviewers r3YY/J8sg]

We believe our work contributes new knowledge and sufficient value to the community, and we hope that the reviewers will reach the same conclusion.

---

### Author Response · Authors · 2025-12-02
**Thanks for the discussion**

We are encouraged that the reviewers were receptive to our response. Before the discussion, the scores were 2, 4, 4, and 6. The reviewers appreciated the additional experimental results provided in the rebuttal, and many of their concerns were addressed. As a result, all 3 reviewers who initially recommended rejection raised their scores. After discussion, the scores are now 4, 6, 6, and 6.

Reviewer GgDp remained cautious about the technical contribution and performance but raised their score from 2 to 4. We appreciate their evaluation but believe the manuscript fully addresses these concerns. Our architecture, a Pairmixer trunk using only triangle multiplication and feed-forward networks paired with a diffusion head, **matches state-of-the-art predictors while offering clear computational benefits across multiple diverse benchmarks**. We also provide **methodological insights** explaining why this simplified design works and will release the model for the community.

Reviewer J8sg seems to agree as they were also initially concerned about novelty, but raised their score from 4 to 6 after seeing our analysis of triangle multiplication. We hope the AC will reach a similar conclusion.

In summary:
* Reviewer qqDT: 4 → 6 (Nov 27)
* Reviewer r3YY: remained at 6 (Nov 25)
* Reviewer GgDp: 2 → 4 (Nov 26)
* Reviewer J8sg: 4 → 6 (Nov 27)

---

### Meta-Review · Area_Chair_obpv · 2026-01-07

**Summary:**

This paper proposes Pairmixer, a simplified AlphaFold3-style trunk that removes triangle attention and sequence updates while retaining triangle multiplication and pairwise feed-forward updates, paired with the same downstream diffusion head. Reviewers broadly agreed that the problem is important and the empirical results are promising, but the initial reviews were not enthusiastic due to two main concerns. First, several reviewers questioned the methodological novelty, arguing that the core observation has been informally known and explored in smaller-scale settings; therefore, the paper needed clearer analysis and framing. Second, reviewers requested broader evidence and empirical testing, including wider benchmark coverage, stronger ablations that isolate which components matter, and a clearer discussion of where the simplification might fail. Following the rebuttal, the additional evaluations and analysis substantially alleviated these concerns, and the discussion trend supports acceptance.

**Reviewer Concerns:**

The request for broader empirical validation was addressed well. The authors conducted diverse evaluations beyond the initial protein-focused setting, which directly addressed the generalization concerns raised in multiple reviews. They also added ablations on a smaller model to isolate the impact of triangle multiplication, triangle attention, and sequence updates, and provided additional diagnostics. Multiple reviewers explicitly acknowledged that these additions addressed most of their concerns, and the score trajectory reflects that.
The request for deeper explanation of why the simplification works addressed. The rebuttal points to a new analysis of triangle multiplication’s role and argues that triangle multiplication can match triangle attention’s functional role in practice, including evidence that it can approximate sparse long-range interactions and that matmul-based implementations yield practical speed gains.

The main remaining issue is novelty. One reviewer remained unconvinced that removing triangle attention constitutes enough methodological contribution for ICLR, even after the additional experiments, and only moved to a borderline score. This suggests that the camera-ready should be careful and concrete in claiming the contribution as a validated, large-scale architectural simplification, along with supporting analysis, rather than overselling the operator itself. A second remaining point is that some comparisons to more conceptually adjacent non-Pairformer architectures were not included, although the rebuttal reasonably argues that these are nontrivial to integrate as fully competitive structure predictors. The paper should clearly state the evaluation scope and why the chosen baselines are the most meaningful for the claim being made.

**Reviewer Scores:**

Overall, the discussion and rebuttal shifted the review set from mixed to supportive, with three reviewers effectively scoring 6 and one scoring 4. Given the strengthened empirical breadth and the added ablations and analysis, I recommend (weak) acceptance.

---

### Decision · Program_Chairs · 2026-01-26

Accept (Poster)